Imperial/TP/21/AB/01

# Discrete gauging and Hasse diagrams

**Guillermo Arias-Tamargo**$^{a,b}$, **Antoine Bourget**$^{c}$, **Alessandro Pini**$^{d}$

$^a$ *Department of Physics, Universidad de Oviedo,*
*C/ Federico García Lorca 18, 33007 Oviedo, Spain*
$^b$ *Instituto Universitario de Ciencias y Tecnologías Espaciales de Asturias (ICTEA)*
*C/ de la Independencia 13, 33004 Oviedo, Spain.*
$^c$ *Theoretical Physics, The Blackett Laboratory, Imperial College SW7 2AZ United Kingdom*
$^d$ *I.N.F.N.- Sezione di Torino, Via Pietro Giuria 1, 10125 Torino, Italy*

*E-mail:* ariasguillermo@uniovi.es, bourget@imperial.ac.uk,
alessandro.pini@to.infn.it

ABSTRACT: We analyse the Higgs branch of 4d $\mathcal{N} = 2$ SQCD gauge theories with non-connected gauge groups $\widetilde{\mathrm{SU}}(N) = \mathrm{SU}(N) \rtimes_{I,II} \mathbb{Z}_2$ whose study was initiated in [1]. We derive the Hasse diagrams corresponding to the Higgs mechanism using adapted characters for representations of non-connected groups. We propose 3d $\mathcal{N} = 4$ magnetic quivers for the Higgs branches in the type $I$ discrete gauging case, in the form of recently introduced wreathed quivers, and provide extensive checks by means of Coulomb branch Hilbert series computations.

## 1 Introduction

Gauge theories play a central role in the current description of high energy physics. The study of gauge theories based on connected and simply connected Lie groups, like $\mathrm{SU}(N)$, has been a very active field of research early on, with the focus being really on the Lie algebra. However it was soon acknowledged that the global structure of the gauge group, beyond its Lie algebra properties, also plays a crucial physical role. The importance of the fundamental group $\pi_1$ has recently been abundantly discussed in the context of supersymmetric gauge theories [2] and the standard model [3].

By contrast, the importance of the group $\pi_0$ of connected components has been less investigated, even though early studies pointed out its physical relevance [4–6], and more recent works connect *finite* gauge groups and dualities in quantum field theory [7, 8]. It should also be mentioned that discrete symmetries appear prominently in the context of higher form symmetries [9], see for instance [10] for a recent account of the class $\mathcal{S}$ case. Another example of the relevance of discrete gaugings is provided by the discovery of new

types of 4d $\mathcal{N} = 3$ SCFTs [11, 12]. These theories are constructed starting with the 4d $\mathcal{N} = 4$ SYM theory with complexified coupling constant $\tau$ tuned to a self dual point of the $\mathrm{SL}(2, \mathbb{Z})$ S-duality group. It turns out that, for these specific values of $\tau$, extra discrete subgroups $\Gamma \subset \mathrm{SL}(2, \mathbb{Z}) \times \mathrm{SU}(4)_R$ are global symmetries of the theories and act in a no trivial way on the supercharges. The gauging of these subgroups breaks the initial amount of supersymmetry down to exactly 12 supercharges, leading this way to 4d $\mathcal{N} = 3$ strongly coupled SCFTs. As discussed in [11, 12], these 4d $\mathcal{N} = 3$ SCFTs are different from those obtained using the S-fold construction [13, 14].

In this article, we are interested in a particular form of discrete symmetry: any simple connected Lie group has a (sometimes trivial) group of outer automorphisms. The automorphisms can intervene in compactification by twisting along cycles, and this can be used to engineer theories with non simply laced gauge groups from string / M theory [15–18]. This is also a much perused tool in F-theory since its early days [19]. The outer automorphisms for a simple complex Lie algebra correspond to the symmetries of its Dynkin diagram, and the the resulting non simply laced algebra is obtained by folding it. In particular it was studied how the Superconformal Index (SCI) [20, 21] of a 4d $\mathcal{N} = 2$ class $\mathcal{S}$ theory is affected by the twist of this symmetry. These theories are obtained starting with the 6d $\mathcal{N} = (2,0)$ theory on $S^3 \times S^1 \times \Sigma_2$ labelled by a simply laced Dynkin diagram $\Gamma$ and performing a compactification over the punctured Riemann surface $\Sigma_2$. In [22] the authors considered the evaluation of the SCI twisted by the outer automorphism group along the $S^1$. Another possibility is to introduce twisted punctures in class $\mathcal{S}$ theories [23–25]; in [26] it was studied how the SCI of type D theories is affected by twist lines on $\Sigma_2$. Similar ideas are considered in [27], where 3d mirror theories of class $\mathcal{S}$ theories of type $A_{2N}$ with twisted punctures compactified on $S^1$ are derived.

Another possibility offered by outer automorphisms is to promote them to gauge symmetries, in effect *extending* the gauge group and making it disconnected. This class of disconnected groups is called *principal extension*, that is to say the disconnected gauge group $\widetilde{G}$ is obtained taking the semidirect product between the connected gauge group $G$ and the discrete outer automorphism group $\Gamma$ of the Dynkin diagram

$$\widetilde{G} \simeq G \rtimes \Gamma. \tag{1.1}$$

As amply discussed below, the equation (1.1) is not sufficient to define the group $\widetilde{G}$: it is necessary to provide an explicit action of $\Gamma$ on $G$. While this construction could seem a bit abstract a well known example is provided by the $\mathrm{O}(2N)$ group that is isomorphic to $\mathrm{SO}(2N) \rtimes \mathbb{Z}_2$, where the discrete group $\mathbb{Z}_2$ acts on the Dynkin diagram of type $D_N$ algebra flipping its two final simple roots. It is then natural to extend the same construction also to the case of type $A_{N-1}$ Lie algebra, that is still endowed with a no trivial $\mathbb{Z}_2$ outer automorphism group. In this case the $\mathbb{Z}_2$ acts on the set of roots $\{\alpha_i\}$ by reflection , i.e. $\alpha_i \leftrightarrow \alpha_{N-i+1}$. The corresponding disconnected group is denoted by $\widetilde{\mathrm{SU}}(N) \simeq \mathrm{SU}(N) \rtimes \mathbb{Z}_2$. From a physical perspective the gauging of this $\mathbb{Z}_2$ corresponds to gauging the charge conjugation symmetry. The study of SCFTs with $\widetilde{\mathrm{SU}}(N)$ gauge groups was initiated in [1] and further extended in [28]. In both these works we focused on a 4d $\mathcal{N} = 2$ context and consider SQCD-like theories, with a $\mathcal{N} = 2$ vector multiplet transforming under the

adjoint representation of $\widetilde{\mathrm{SU}}(N)$ and matter provided by $\mathcal{N} = 2$ hypermultiplets in the fundamental representation of the gauge group.

In the discretely gauged theory with $\widetilde{\mathrm{SU}}(N)$ gauge group the gauge and the matter fields transform under representations of the disconnected gauge group. This is the place in which the different global structures of the groups play a crucial role since, in general, the representations of $\widetilde{\mathrm{SU}}(N)$ differ from representations of $\mathrm{SU}(N)$. Moreover it was observed in [28] that when $N$ is even there are two non equivalent ways of performing the gauging of the $\mathbb{Z}_2$ symmetry, that give rise to two distinct gauge groups, that have been denoted by $\widetilde{\mathrm{SU}}(N)_I$ and $\widetilde{\mathrm{SU}}(N)_{II}$ respectively. On the other hand, when $N$ is odd, there is only one possibility corresponding to $\widetilde{\mathrm{SU}}(N)_I$. From a mathematical point of view these two possibilities, arising in the $N$ even case, are related to the fact that the complexified Lie algebra $\mathfrak{sl}(N,\mathbb{C})$ admits two distinct real forms that give rise to two non equivalent ways of gauging charge conjugation.[1]

All the theories that we study are endowed with a moduli space of vacua parametrized by BPS chiral scalar gauge invariant operators. It is then natural to investigate how the discrete gauging action affects these spaces. From a physical point of view the fact that the gauge group has become larger introduces further restrictions on the types of gauge invariant operators that we can construct and therefore, we expect a modification of the geometric structure of the corresponding moduli space. A systematic way to characterize the geometry of these moduli spaces is provided by the *Plethystic program* [29, 30], with the central notion of Hilbert series, a generating function that counts the chiral operators present in the theory according to their conformal dimension and other quantum numbers [31, 32]. The extension of these tools, in the context of principal extensions, was performed in [33] and we employ them in our analysis.

Moreover, even if the complete characterization the full moduli space of vacua is in general very difficult, for a 4d $\mathcal{N} = 2$ gauge theory we can identify two particular sub-branches, namely the *Coulomb branch* and *Higgs branch*. Specifically the Coulomb branch arises when we give a vacuum expectation value (VEV) to the complexified scalar inside the $\mathcal{N} = 2$ vector multiplet. For the theories discussed in this article the computation of the Hilbert series of the Coulomb branch was performed in [1, 28]. Remarkably it was found that the Coulomb branch of these theories is not freely generated. On the other hand the Higgs branch is parameterized by the VEVs of the scalar fields inside the $\mathcal{N} = 2$ hypermultiplets. In general if there is enough matter in the theory a generic VEV completely breaks the gauge group. Nevertheless we can also give a VEV only to a subset of the scalar fields, this way the gauge group could be broken to a no-trivial subgroup. This partial Higgs mechanism is naturally described by a partial order diagram, called the *Hasse diagram*, where each node of the diagram is related to the subgroup of the initial gauge group that is left unbroken by the Higgs mechanism. The systematic study of the Higgs branch of theories with 8 supercharges using Hasse diagrams was initiated in [34] and further analysed in [35–45]. The Higgs branch Hasse diagram in turns reveals the

---

[1]See Section 2.3 of [28] for details on that point, and Appendix A for a compendium of the essential definitions.

geometric structure of the Higgs branch as a symplectic singularity, the nodes being in correspondence with symplectic leaves, and the links representing elementary transverse slices. In this paper we aim to move a further step in this direction and we analyse how the structure of the Higgs branch of the SQCD-like theories with $\widetilde{\mathrm{SU}}(N)_I$ or $\widetilde{\mathrm{SU}}(N)_{II}$ gauge groups is revealed by the partial Higgsing procedure described above. In particular our first main result is the derivation of the Hasse diagrams for Type I and Type II gauging in Figure 3 and Figure 5. This is based on a careful analysis of representations of $\widetilde{\mathrm{SU}}(N)_{I/II}$ groups, their characters and branching rules.

The Higgs branch of certain 4d $\mathcal{N} = 2$ theories can be equivalently described as the Coulomb branch of 3d $\mathcal{N} = 4$ quiver gauge theories. When this is the case, the quiver is called a *magnetic quiver* for that Higgs branch [46–50]. Our second main result is a magnetic quiver for the Higgs branch of $\widetilde{\mathrm{SU}}(N)_I$ theories, in the form of a *wreathed quiver*, as introduced in [39]. As a check of our conjecture we compute the 3d $\mathcal{N} = 4$ Coulomb branch Hilbert series of that quiver and find perfect agreement with the Higgs branch Hilbert series of the corresponding $\widetilde{\mathrm{SU}}(N)_I$ theory that was computed in [1, 28]. The computation is performed using the *monopole formula* originally introduced in [31] and generalized to wreathed quivers in [39].

The present article is organized as follows. In Section 2 we introduce the notion of characters for representations of disconnected groups and we discuss the derivation of the branching rules relevant for the partial Higgsing mechanisms discussed in this article. In Section 3 we briefly review the notion of Hasse diagram and we discuss its construction for type I and type II discretely gauged theories. In Section 4 we review the generalization of the monopole formula in the context of 3d $\mathcal{N} = 4$ wreathed quiver gauge theories and we apply it to theories of type I providing a candidate magnetic quiver. The appendices gather basic definitions and technicalities regarding $\widetilde{\mathrm{SU}}(N)$ groups.

## 2 Characters and branching rules for disconnected groups

In this section we develop tools that allow to use the theory of characters of Lie groups in the context of disconnected groups, focusing on the examples of $\mathrm{O}(N)$ and $\widetilde{\mathrm{SU}}(N)$. This allows to compute tensor products, and more importantly branching rules, which are needed to compute Hasse diagrams in the next section.

Writing the characters for a group $G$ (connected or not) requires firstly the identification of irreducible representations $\rho : G \to \mathrm{GL}(V)$, and secondly the choice of a subgroup $T \subset G$ parametrized by fugacities (which can assume continuous or discrete values). The character is then the function $\chi_\rho : T \to \mathbb{C}$ defined by $\chi_\rho(t) = \mathrm{Tr}(\rho(t))$ for $t \in T$. The new feature of this analysis for disconnected groups $G$ is the appearance of discrete fugacities in $T$. This can be seen as a fusion between the usual theories of characters of connected Lie group on one side, and of representation theory of finite groups (here the component group of $G$) on the other side. Here we consider only the simplest non trivial case (1.1)

where $\Gamma = \mathbb{Z}_2$, which has character table

$$\begin{array}{c|cc} \epsilon & 1 & -1 \\ \hline \chi_{\mathbf{1}} & 1 & 1 \\ \chi_\epsilon & 1 & -1 \end{array} \tag{2.1}$$

but the principles would remain valid for a larger component group. In the character table (2.1), the two $\mathbb{Z}_2$ elements are denoted by $\epsilon = \pm 1$. and rows of this table contain the characters of its two irreducible representations.[2]

## 2.1 Representations and characters for $O(N)$

**Groups $O(2N)$**

We start with the very simple example of O(2) to setup the concepts and notations in a framework where everything can be written explicitly. This group is a semidirect product $SO(2) \rtimes \mathbb{Z}_2$, so an element of O(2) can be written as a pair $(g, \epsilon) \in SO(2) \times \mathbb{Z}_2$. The semidirect product is specified by the $\Theta_\epsilon$ automorphism of SO(2) defined by

$$\Theta_1 \begin{pmatrix} \cos\theta & -\sin\theta \\ \sin\theta & \cos\theta \end{pmatrix} = \begin{pmatrix} \cos\theta & -\sin\theta \\ \sin\theta & \cos\theta \end{pmatrix}, \qquad \Theta_{-1} \begin{pmatrix} \cos\theta & -\sin\theta \\ \sin\theta & \cos\theta \end{pmatrix} = \begin{pmatrix} \cos\theta & \sin\theta \\ -\sin\theta & \cos\theta \end{pmatrix} . \tag{2.2}$$

Note that $\Theta_{-1}$ is the conjugation by the reflection matrix $\mathrm{Diag}(-1, 1)$. The fundamental representation is

$$O(2) = \left\{ \begin{pmatrix} \cos\theta & -\sin\theta \\ \sin\theta & \cos\theta \end{pmatrix} \middle| \theta \in T^1 \right\} \cup \left\{ \begin{pmatrix} \cos\theta & \sin\theta \\ \sin\theta & -\cos\theta \end{pmatrix} \middle| \theta \in T^1 \right\} . \tag{2.3}$$

Setting $z = e^{i\theta}$, the trace of the matrices in the identity component is $z + z^{-1}$ while the trace vanishes in the disconnected component. Therefore the character can be written as a function of $z$ and $\epsilon$ as

$$\chi^{O(2)}_{\text{Fundamental}}(z, \epsilon) = \left( \frac{1 + \epsilon}{2} \right) (z + z^{-1}) = \begin{cases} z + z^{-1} & \text{if } \epsilon = 1 \\ 0 & \text{if } \epsilon = -1 \end{cases} . \tag{2.4}$$

The character has two fugacities, one continuous variable $z$ and one discrete variable $\epsilon$, and they span the *fugacity group*.

Consider now the adjoint representation, i.e. the action of $(g, \epsilon) \in O(2)$ on $a \in \mathbb{R}$ given by

$$\begin{pmatrix} 0 & a \\ -a & 0 \end{pmatrix} \mapsto (g, \epsilon) \begin{pmatrix} 0 & a \\ -a & 0 \end{pmatrix} (g, \epsilon)^{-1} . \tag{2.5}$$

This is $a \mapsto a$ for $\epsilon = 1$ and $a \mapsto -a$ for $\epsilon = -1$. Therefore the corresponding character reads

$$\chi^{O(2)}_{\text{Adjoint}}(z, \epsilon) = \left( \frac{1 + \epsilon}{2} \right) \times (1) + \left( \frac{1 - \epsilon}{2} \right) \times (-1) = \epsilon . \tag{2.6}$$

---

[2]We slightly abuse notation in denoting by the same symbol $\epsilon$ two related objects, namely the generic element of $\mathbb{Z}_2$ (which plays the role of a discrete fugacity, satisfying $\epsilon^2 = 1$) and the non-trivial irreducible representation of $\mathbb{Z}_2$. With this choice the character of the $\epsilon$ representation is $\epsilon$.

We note an interesting fact: the adjoint representation is not the same as the trivial representation. There are two inequivalent representations of dimension 1, while there is only one of dimension 2.

Consider now $O(4) = SO(4) \rtimes \mathbb{Z}_2$, the first example in which the choice of the subgroup of fugacities $T$ is not straightforward. As maximal torus of $SO(4)$ we choose matrices of the form

$$
\begin{pmatrix}
\cos\theta_1 & -\sin\theta_1 & 0 & 0 \\
\sin\theta_1 & \cos\theta_1 & 0 & 0 \\
0 & 0 & \cos\theta_2 & -\sin\theta_2 \\
0 & 0 & \sin\theta_2 & \cos\theta_2
\end{pmatrix}
\tag{2.7}
$$

The trace of this matrix is $z_1 + z_1^{-1} + z_2 + z_2^{-1}$ with $z_1 = e^{i\theta_1}$ and $z_2 = e^{i\theta_2}$. However we now need to specify how the semidirect product is defined, as there is no way to make this choice symmetric in $z_1$ and $z_2$. We *choose* to define $\Theta_{-1}$ as the conjugation by the reflection $\mathrm{Diag}(-1,1,1,1)$. As a consequence, the trace of an element with $\epsilon = -1$ in the fundamental representation is $z_2 + z_2^{-1}$. The symmetry between $z_1$ and $z_2$ is broken. The embedding $O(2) \subset O(4)$ is obtained by sending $z_2 \to 1$, while sending $z_1 \to 1$ gives the embedding $SO(2) \subset O(4)$. The reader is encouraged to check this on the characters of the fundamental and adjoint representations of $O(4)$ which read

$$
\chi^{O(4)}_{\mathrm{Fundamental}}(z_1, z_2, \epsilon) = \left(\frac{1+\epsilon}{2}\right)(z_1 + z_1^{-1}) + (z_2 + z_2^{-1}) \, ,
\tag{2.8}
$$

$$
\chi^{O(4)}_{\mathrm{Adjoint}}(z_1, z_2, \epsilon) = \left(\frac{1+\epsilon}{2}\right)\left(2 + (z_1 + z_1^{-1})(z_2 + z_2^{-1})\right) \, .
\tag{2.9}
$$

One can generalize these computations to $O(2N) = SO(2N) \rtimes \mathbb{Z}_2$, with $\Theta_{-1}$ given by conjugation by $\mathrm{Diag}(-1, 1, \cdots, 1)$. The characters of the fundamental and adjoint representations of $O(2N)$ are

$$
\chi^{O(2N)}_{\mathrm{Fundamental}}(z_i, \epsilon) = \left(\frac{1+\epsilon}{2}\right)(z_1 + z_1^{-1}) + \sum_{i=2}^{N}(z_i + z_i^{-1}) \, ,
\tag{2.10}
$$

$$
\chi^{O(2N)}_{\mathrm{Adjoint}}(z_i, \epsilon) = \left(\frac{1+\epsilon}{2}\right)\left(2 + (z_1 + z_1^{-1})\sum_{2 \leq j \leq N}(z_j + z_j^{-1})\right)
$$
$$
+ (N-2) + \sum_{2 \leq i < j \leq N}(z_i + z_i^{-1})(z_j + z_j^{-1}) \, .
\tag{2.11}
$$

Note that the group $O(2N)$ is simple for $N \geq 3$, and in those cases the trivial, fundamental and adjoint representations are given by Dynkin labels $[0, \ldots, 0]$, $[1, 0, \ldots, 0]$ and $[0, 1, 0, \ldots, 0]$ respectively. These are invariant under the exchange of the Dynkin labels for the two spinor nodes, so from each of these representation one can build another inequivalent one by tensoring with $\epsilon$.[3] In the case $N = 2$, the group $O(2N)$ is not simple, and accordingly the adjoint representation corresponds to Dynkin labels $[2, 0] \oplus [0, 2]$; from the

---

[3]These representations are sometimes called "pseudo". From the trivial or scalar representation one builds the pseudo-scalar, and from the fundamental or vector one builds the pseudo-vector.

| $\mathrm{O}(2N)$ | $\longrightarrow$ | $\mathrm{O}(2N-1) \times \mathrm{O}(1) = (\mathrm{SO}(2N-1) \times \mathbb{Z}_2) \times \mathbb{Z}_2$ |
|---|---|---|
| $F_{\mathrm{O}(2N)}$ | $\longmapsto$ | $\left(F_{\mathrm{SO}(2N-1)} \otimes \epsilon \otimes \mathbf{1}\right) \oplus \left(\mathbf{1}_{\mathrm{SO}(2N-1)} \otimes \mathbf{1} \otimes \epsilon\right)$ |
| $\mathrm{Adj}_{\mathrm{O}(2N)}$ | $\longmapsto$ | $\left(\mathrm{Adj}_{\mathrm{SO}(2N-1)} \otimes \mathbf{1} \otimes \mathbf{1}\right) \oplus \left(F_{\mathrm{SO}(2N-1)} \otimes \epsilon \otimes \epsilon\right)$ |

| $\mathrm{SO}(2N+1)$ | $\longrightarrow$ | $\mathrm{O}(2N)$ |
|---|---|---|
| $F_{\mathrm{SO}(2N+1)}$ | $\longmapsto$ | $F_{\mathrm{O}(2N)} \oplus \mathbf{1}_{\mathrm{O}(2N)}$ |
| $\mathrm{Adj}_{\mathrm{SO}(2N+1)}$ | $\longmapsto$ | $\mathrm{Adj}_{\mathrm{O}(2N)} \oplus F_{\mathrm{O}(2N)}$ |

**Table 1**. Summary of branching rules for $\mathrm{O}(N)$ groups.

general arguments given in [28, Sec. 3.1], it follows that the character should vanish for $\epsilon = -1$, and this is indeed the case in (2.11).

**Groups $O(2N+1)$ and Branching rules**

The group $\mathrm{O}(2N+1)$ is a direct product $\mathrm{SO}(2N+1) \times \mathbb{Z}_2$ so the characters factorize. Using the characters one can check the branching rules for orthogonal groups. The branching rules $\mathrm{O}(2N+1) \to \mathrm{O}(2N)$ are obtained by restricting the $\mathrm{O}(2N+1)$ characters to $\mathrm{O}(2N)$ characters, with no change in the fugacities. The branching rules $\mathrm{O}(2N) \to \mathrm{O}(2N-1)$ are obtained by setting $z_N \to 1$. The results are presented in Table 1.

## 2.2 Representations and characters for $\widetilde{\mathrm{SU}}(N)$

We can apply similar techniques to express characters of representations of $\widetilde{\mathrm{SU}}(N)_{I/II}$. Definitions and notations for these groups are gathered in Appendix A. As for $\mathrm{O}(N)$, we first have two one-dimensional representations:

1. The trivial representation, with character $\chi_1^{\widetilde{\mathrm{SU}}(N)} = 1$.

2. The $\epsilon$ representation, with character $\chi_\epsilon^{\widetilde{\mathrm{SU}}(N)} = \epsilon$.

Let us now move to higher dimensional representations. As explained in [28], representations of $\mathrm{SU}(N)$ induce representations of $\widetilde{\mathrm{SU}}(N)$ according to the following rule. Let $R$ be a representation of $\mathrm{SU}(N)$ with highest weight $\lambda$. If $\lambda = [\lambda_1, \ldots, \lambda_{N-1}]$ is invariant under the permutation $\lambda_i \leftrightarrow \lambda_{N-i}$ then there are two corresponding representations of $\widetilde{\mathrm{SU}}(N)$, both of dimension $\dim(R)$, which differ by a tensor product with the $\epsilon$ representation; if the highest weight is not invariant under that permutation, then there is a single corresponding representation of $\widetilde{\mathrm{SU}}(N)$, of dimension $2\dim(R)$.

In this paper we focus on the representations induced by the fundamental and the adjoint of $\mathrm{SU}(N)$. These are

| Representation | Value on $(g,1)$ | Value on $(g,-1)$ |
|---|---|---|
| Trivial | $1$ | $1$ |
| $\epsilon$ | $1$ | $-1$ |
| $F \oplus \overline{F}$ | $\begin{pmatrix} g & 0 \\ 0 & \Theta_{-1}(g) \end{pmatrix}$ | $\begin{pmatrix} 0 & g \\ \Theta_{-1}(g) & 0 \end{pmatrix}$ |
| Adj | $X \mapsto gXg^{-1}$ | $X \mapsto g\theta_{-1}(X)g^{-1}$ |
| Adj $\otimes\,\epsilon$ | $X \mapsto gXg^{-1}$ | $X \mapsto -g\theta_{-1}(X)g^{-1}$ |

| Representation | Character |
|---|---|
| Trivial | $1$ |
| $\epsilon$ | $\epsilon$ |
| $F \oplus \overline{F}$ | $\left(\frac{1+\epsilon}{2}\right) \sum_{i=0}^{N-1} \left( \frac{z_i}{z_{i+1}} + \frac{z_{i+1}}{z_i} \right)$ |
| Adj | $\begin{cases} \left(\frac{1+\epsilon}{2}\right)\left(-1 + \sum_{i=0}^{N-1}\sum_{j=0}^{N-1} \frac{z_i}{z_{i+1}}\frac{z_{j+1}}{z_j}\right) + (1-N)\left(\frac{1-\epsilon}{2}\right) & \text{Type } I \\[2em] \left(\frac{1+\epsilon}{2}\right)\left(-1 + \sum_{i=0}^{N-1}\sum_{j=0}^{N-1} \frac{z_i}{z_{i+1}}\frac{z_{j+1}}{z_j}\right) + (1+N)\left(\frac{1-\epsilon}{2}\right) & \text{Type } II \end{cases}$ |
| Adj $\otimes\,\epsilon$ | $\begin{cases} \left(\frac{1+\epsilon}{2}\right)\left(-1 + \sum_{i=0}^{N-1}\sum_{j=0}^{N-1} \frac{z_i}{z_{i+1}}\frac{z_{j+1}}{z_j}\right) - (1-N)\left(\frac{1-\epsilon}{2}\right) & \text{Type } I \\[2em] \left(\frac{1+\epsilon}{2}\right)\left(-1 + \sum_{i=0}^{N-1}\sum_{j=0}^{N-1} \frac{z_i}{z_{i+1}}\frac{z_{j+1}}{z_j}\right) - (1+N)\left(\frac{1-\epsilon}{2}\right) & \text{Type } II \end{cases}$ |

**Table 2**. The first table shows the representations of $\widetilde{\mathrm{SU}}(N)$ used in the paper, by giving the explicit action of elements of the form $(g,\epsilon)$ for $\epsilon = \pm 1$. For the 1-dimensional representations, this is a number; for the $F \oplus \overline{F}$ representation we give a $2N \times 2N$ matrix, and for the adjoint and $\epsilon$-adjoint we give the action on an element $X$ in the Lie algebra $\mathfrak{g}$. The second table gives the characters expressed in terms of fugacities $(z_1, \ldots, z_{N-1}, \epsilon) \in \mathcal{T}$ with the convention $z_0 = z_N = 1$.

3. The fundamental representation. This is a $2N$ dimensional representation which we denote by $(F \oplus \overline{F})$. Let us emphasize that despite this notation, this is an *irreducible* representation, as the $\mathbb{Z}_2$ element mixes the $F$ and $\overline{F}$ of the starting $\mathrm{SU}(N)$ group.

4. The adjoint representation, of dimension $N^2 - 1$.

5. The tensor product of the adjoint representation with $\epsilon$ representation, of dimension $N^2 - 1$.

To write down characters for these representations, we need to pick a group of fugacities. For characters in $\mathrm{SU}(N)$, the natural choice is to consider the subgroup of diagonal matrices $\mathrm{U}(1)^{N-1}$. In the case of the disconnected group $\widetilde{\mathrm{SU}}(N)$, it turns out that the choice of an appropriate fugacity subgroup is a subtle problem that is discussed at length in Appendix A. In a nutshell, the reason for which the choice is subtle is that certain subgroups,

| $\widetilde{\mathrm{SU}}(N)_I$ | $\longrightarrow$ | $\widetilde{\mathrm{SU}}(N-1)_I$ |
|---|---|---|
| $(F \oplus \overline{F})_{\widetilde{\mathrm{SU}}(N)_I}$ | $\longmapsto$ | $(F \oplus \overline{F})_{\widetilde{\mathrm{SU}}(N-1)_I} \oplus \epsilon \oplus \mathbf{1}$ |
| $\mathrm{Adj}_{\widetilde{\mathrm{SU}}(N)_I}$ | $\longmapsto$ | $\mathrm{Adj}_{\widetilde{\mathrm{SU}}(N-1)_I} \oplus (F \oplus \overline{F})_{\widetilde{\mathrm{SU}}(N-1)_I} \oplus \epsilon$ |
| $\epsilon$ | $\longmapsto$ | $\epsilon$ |

| $\widetilde{\mathrm{SU}}(2N)_{II}$ | $\longrightarrow$ | $\widetilde{\mathrm{SU}}(2N-2)_{II}$ |
|---|---|---|
| $(F \oplus \overline{F})_{\widetilde{\mathrm{SU}}(2N)_{II}}$ | $\longmapsto$ | $(F \oplus \overline{F})_{\widetilde{\mathrm{SU}}(2N-2)_{II}} \oplus 2 \times \epsilon \oplus 2 \times \mathbf{1}$ |
| $\mathrm{Adj}_{\widetilde{\mathrm{SU}}(2N)_{II}}$ | $\longmapsto$ | $\mathrm{Adj}_{\widetilde{\mathrm{SU}}(2N-2)_{II}} \oplus 2 \times (F \oplus \overline{F})_{\widetilde{\mathrm{SU}}(2N-2)_{II}} \oplus \epsilon \oplus 3 \times \mathbf{1}$ |
| $\epsilon$ | $\longmapsto$ | $\epsilon$ |

| $\widetilde{\mathrm{SU}}(N)_{I,II}$ | $\longrightarrow$ | $\mathrm{SU}(N)$ |
|---|---|---|
| $(F \oplus \overline{F})_{\widetilde{\mathrm{SU}}(N)}$ | $\longmapsto$ | $F_{\mathrm{SU}(N)} \oplus \overline{F}_{\mathrm{SU}(N)}$ |
| $\mathrm{Adj}_{\widetilde{\mathrm{SU}}(N)}$ | $\longmapsto$ | $\mathrm{Adj}_{\mathrm{SU}(N)}$ |
| $\epsilon$ | $\longmapsto$ | $\mathbf{1}$ |

**Table 3**. Summary of branching rules for $\widetilde{\mathrm{SU}}(N)$ groups.

called *Cartan subgroups*, are well suited for representation theory (e.g. an extension of the Weyl character formula is valid) but do not commute with the embedding of smaller disconnected groups like $\widetilde{\mathrm{SU}}(N-1) \subset \widetilde{\mathrm{SU}}(N)$. In the present section, we are interested in branching rules for that embedding, so we pick instead $\mathcal{T} = \{(z_1, \ldots, z_{N-1}, \epsilon)\} = \mathrm{U}(1)^{N-1} \rtimes \mathbb{Z}_2$ as defined in (A.23), where the first factor corresponds to diagonal matrices in $\mathrm{SU}(N)$ and the semidirect product is the one that serves to define the extension $\widetilde{\mathrm{SU}}(N)$. With these choices, the representations and their characters are summarized in Table 2.

For instance, in the fundamental representation the trace of the matrix corresponding to an element $(g, -1)$ is clearly 0, so that the character is the product of the the corresponding $\mathrm{SU}(N)$ character by the projector $\frac{1+\epsilon}{2}$. As another example, for the adjoint representation the action of $(g, -1)$ on the Lie algebra is $X \mapsto g\theta_{-1}(X)g^{-1}$, see equation (A.13). As shown in section A.3, the computation of the character reduces to the computation of the trace of $\theta_{-1}$, which is given in equations (A.14) and (A.15).

The characters of Table 2 allow to compute branching rules. For instance the branching rules for fundamentals are

$$\chi_{(F \oplus \overline{F})_{\widetilde{\mathrm{SU}}(N)}}\big|_{z_{N-1} \to 1} = \left(\frac{1+\epsilon}{2}\right) \sum_{i=0}^{N-2} \left(\frac{z_i}{z_{i+1}} + \frac{z_{i+1}}{z_i}\right) + 2\left(\frac{1+\epsilon}{2}\right)$$
$$= \chi_{(F \oplus \overline{F})_{\widetilde{\mathrm{SU}}(N-1)}} + 1 + \epsilon. \tag{2.12}$$

For a less trivial example, let us look at the adjoint representation in type *II*. We take

$N \geq 4$ even and consider the branching rules for the embedding $\widetilde{\mathrm{SU}}(N-2)_{II} \subset \widetilde{\mathrm{SU}}(N)_{II}$

$$
\begin{aligned}
\chi_{\mathrm{Adj}_{\widetilde{\mathrm{SU}}(N)_{II}}}|_{z_{N-2}, z_{N-1} \to 1} &= \left(\frac{1+\epsilon}{2}\right)\left(\chi_{\mathrm{Adj}_{\mathrm{SU}(N)}}|_{z_{N-2}, z_{N-1} \to 1}\right) + (1+N)\left(\frac{1-\epsilon}{2}\right) \\
&= \left(\frac{1+\epsilon}{2}\right)\left(\chi_{\mathrm{Adj}_{\mathrm{SU}(N-2)}} + 2\chi_{(F \oplus \overline{F})_{\mathrm{SU}(N-2)}} + 4\right) + (1+N)\left(\frac{1-\epsilon}{2}\right) \\
&= \chi_{\mathrm{Adj}_{\widetilde{\mathrm{SU}}(N-2)_{II}}} + 2\chi_{(F \oplus \overline{F})_{\widetilde{\mathrm{SU}}(N-2)_{II}}} + 3 + \epsilon \,.
\end{aligned}
\tag{2.13}
$$

The crucial feature here is the $3 + \epsilon$ contribution. If instead we repeat the computation for type $I$ this term becomes $1 + 3\epsilon$ because of the different sign in front of the $N\epsilon$ term in the character. The branching rules are summarized in Table 3.

# 3 Hasse diagrams

Higgs branches of theories with 8 supercharges are hyperKähler cones [51], or symplectic singularities [52], and as such admit a foliation [53, 54] which can be conveniently described by a Hasse diagram. Each point of the diagram represents a symplectic leaf of said foliation. The Hasse diagram represents a partial order between the symplectic leaves, defined by inclusions in their closures. For any two given leaves which can be compared in this partial order, we have a transverse slice which describes how the smaller leaf looks as a symplectic singularity inside the closure of the bigger leaf. If the two leaves are adjacent in the Hasse diagram, we have a so called elementary transverse slice, which oftentimes has a simple geometric description as the closure of a minimal nilpotent orbit of a classical group or as a Klein singularity (this will always be the case for our purposes, see [44, 55] for examples with more exotic elementary slices).

In [34] the Hasse diagram of symplectic leaves for the Higgs branch of a classical gauge theory with 8 supercharges is identified with the Hasse diagram of phases of that gauge theory under partial Higgsing. Each leaf is labeled by the unbroken gauge group in that phase, and the elementary transverse slices describe the geometry of gauge singlets. We apply this principle to the Higgs branch of gauge theories with $\widetilde{\mathrm{SU}}$ gauge groups, and in this section, we derive the Hasse diagrams of Figures 1, 3 and 5 by looking at the chain of possible Higgsing patterns. As a warm-up we first review that procedure by looking at the example of $\mathrm{SU}(N_c) + N_f$ SQCD.

The Higgs branch of this theory is defined classically as a hyperKähler quotient which can be written symbolically as

$$
\frac{1}{2}\left(N_f F_{N_c} + N_f \overline{F}_{N_c}\right) - \mathrm{Adj}_{N_c} \,,
\tag{3.1}
$$

where the factor of $1/2$ is due to the fact that when separating fundamentals and antifundamentals we are counting half-hypers. The hyperKähler quotient is denoted by the minus sign in the above equation. Replacing each representation by its dimension, the formula above yields the quaternionic dimension of the Higgs branch.

| Slice | $\dim_{\mathbb{H}}$ | Gauge Theory | Global Symmetry |
|-------|---------------------|--------------|-----------------|
| $a_N$ | $N$ | U(1) with $N+1$ fundamental hypermultiplets | $\mathfrak{su}(N+1)$ |
| $c_N$ | $N$ | $O(1) = \mathbb{Z}_2$ with $2N$ fundamental half-hypermultiplets | $\mathfrak{sp}(N)$ |
| $d_N$ | $2N-3$ | Sp(1) with $2N$ fundamental half-hypermultiplets | $\mathfrak{so}(2N)$ |

**Table 4**. List of elementary slices which appear in this paper. These are the closures of the minimal nilpotent orbits of their global symmetry algebra.

Now we apply the branching rules under the breaking $SU(N_c) \to SU(N_c - 1)$,

$$\frac{N_f}{2}\left[F_{N_c-1} \oplus \mathbf{1}_{N_c-1}\right] + \frac{N_f}{2}\left[\overline{F}_{N_c-1} \oplus \mathbf{1}_{N_c-1}\right] - \left[\text{Adj}_{N_c-1} \oplus F_{N_c-1} \oplus \overline{F}_{N_c-1} \oplus \mathbf{1}_{N_c-1}\right] . \tag{3.2}$$

Finally, we reshuffle this expression to put it in the form

$$[\text{Matter fields charged under } SU(N_c - 1)] - [\text{Adjoint of } SU(N_c - 1)] + [\text{singlets}] . \tag{3.3}$$

The first two terms identify the theory that results after the Higgsing, and the singlets correspond to the transverse slice according to Table 4. The cancellation of the fundamentals coming from $\text{Adj}_{N_c}$ by fundamentals coming from matter fields corresponds physically to the Higgs mechanism, where some of the gauge bosons of the initial theory acquire a mass. In our example,

$$\frac{1}{2}\left((N_f - 2)F_{N_c-1} + (N_f - 2)\overline{F}_{N_c-1}\right) - \text{Adj}_{N_c-1} \oplus \left[\frac{1}{2}(N_f + N_f) - 1\right]\mathbf{1}_{N_c-1}, \tag{3.4}$$

which means that the remaining theory after Higgsing is $SU(N_c - 1) + (N_f - 2)$ SQCD, and the transverse slice is $a_{N_f-1}$ i.e. the (closure of the) minimal nilpotent orbit of $\mathfrak{su}(N_f)$. The slice is identified to be $a_{N_f-1}$ as the coefficient of the singlets of the hyperKähler quotient of the U(1) gauge theory as described in Table 4.

## 3.1 Hasse Diagram for O

Let's proceed with our first disconnected group, and consider a theory with gauge group $O(N_c)$ plus $N_f \geq N_c$ fields in the fundamental representation. The Hasse diagram is already shown in [34]; we rederive it here to illustrate the method of characters for disconnected groups. We use the same procedure shown above to find the transverse slices and resulting theories after the Higgsing. In order to make sure that we get the full Hasse diagram, we need to scan over the possible subgroups of $O(N_c)$ and check which symmetry breaking patterns are possible according to the branching rules of Table 1. Let's begin by considering the (potential) breaking $O(N_c) \to O(N_c - 1) \times O(1)$. Note that since $O(2k + 1)$ can be written as a direct product, but $O(2k)$ cannot, there is in principle a difference between choosing $N_c$ odd or even. We take $N_c$ even for now, and shall soon see that this initial choice doesn't matter.

$$N_f F_{O(N_c)} - \text{Adj}_{O(N_c)} \to N_f[(F_{SO(N_c-1)} \otimes \epsilon \otimes \mathbf{1}) \oplus (\mathbf{1}_{SO(N_c-1)} \otimes \mathbf{1} \otimes \epsilon)]$$
$$- [(\text{Adj}_{SO(N_c-1)} \otimes \mathbf{1} \otimes \mathbf{1}) \oplus (F_{SO(N_c-1)} \otimes \epsilon \otimes \epsilon)]. \tag{3.5}$$

Note that the last term coming from the decomposition of the adjoint cannot be cancelled. This means that under the symmetry breaking pattern under consideration, the gauge fields have no Goldstone bosons to eat, and therefore the Higgs mechanism cannot take place. In a similar way, we can check that $O(N_c)$ also can't break to the subgroup $O(p) \times O(q)$ (with $p + q = N_c$).

The next possible breaking to consider is then $O(N_c) \to O(N_c - 1)$. This is achieved taking (3.5) and forgetting the second $\mathbb{Z}_2$ representation in each tensor product, as that was the one corresponding to the $O(1)$ factor. Thus we have

$$
\begin{aligned}
N_f F_{O(N_c)} - \mathrm{Adj}_{O(N_c)} \to & N_f[(F_{SO(N_c-1)} \otimes \epsilon) \oplus (\mathbf{1}_{SO(N_c-1)} \otimes \mathbf{1})] \\
& - [(\mathrm{Adj}_{SO(N_c-1)} \otimes \mathbf{1}) \oplus (F_{SO(N_c-1)} \otimes \epsilon)].
\end{aligned} \tag{3.6}
$$

We see that this symmetry breaking pattern is possible, and it results in

$$
N_f F_{O(N_c)} - \mathrm{Adj}_{O(N_c)} \to (N_f - 1) F_{O(N_c-1)} - \mathrm{Adj}_{O(N_c-1)} \underbrace{\oplus N_f \mathbf{1}_{O(N_c-1)}}_{c_{N_f} \text{ slice}}. \tag{3.7}
$$

From this, we conclude that the transverse slice at the bottom of the Hasse diagram is $c_{N_f}$, and the remaining theory on the symplectic leaf of the Higgs branch is $O(N_c - 1) + (N_f - 1)F$. Recall that we had chosen $N_c$ even, so now we have an odd number of colours, $O(N_c - 1) = \mathbb{Z}_2 \times SO(N_c - 1)$. We can therefore consider the potential breaking $\mathbb{Z}_2 \times SO(N_c - 1) \to \mathbb{Z}_2 \times O(N_c - 2)$, where the $\mathbb{Z}_2$ representations stay the same and the branching rules are those in the second part of Table 1. This results in

$$
\begin{aligned}
(N_f - 1)[\epsilon \otimes F_{SO(N_c-1)}] - [\mathbf{1} \otimes \mathrm{Adj}_{SO(N_c-1)}] \to & (N_f - 1)[\epsilon \otimes (F_{O(N_c-2)} \oplus \mathbf{1}_{O(N_c-2)})] \\
& - [\mathbf{1} \otimes (\mathrm{Adj}_{O(N_c-2)} \oplus F_{O(N_c-2)})] \\
= & (N_f - 1)[(\epsilon \otimes F_{O(N_c-2)}) \oplus (\epsilon \otimes \mathbf{1}_{O(N_c-2)})] \\
& - [(\mathbf{1} \otimes \mathrm{Adj}_{O(N_c-2)}) \oplus (\mathbf{1} \otimes F_{O(N_c-2)})].
\end{aligned} \tag{3.8}
$$
$$
\tag{3.9}
$$

Once again, we see that the necesary cancellations are only possible if we forget about the $\mathbb{Z}_2$, i.e. if we consider the breaking $O(N_c - 1) \to O(N_c - 2)$. With this,

$$
(N_f - 1) F_{O(N_c-1)} - \mathrm{Adj}_{O(N_c-1)} \to (N_f - 2) F_{O(N_c-2)} - \mathrm{Adj}_{O(N_c-2)} \underbrace{\oplus (N_f - 1) \mathbf{1}_{O(N_c-2)}}_{c_{N_f-1} \text{ slice}}. \tag{3.10}
$$

That is, the second slice at the bottom of the Hasse diagram is $c_{N_f-1}$ and the remaining theory is $O(N_c - 2) + (N_f - 2)F$. To complete the Hasse diagram, we need to repeat the process above the necessary number of times. Note that, as advertised, it doesn't matter whether at each step the number of colour is even or odd; even if the decomposition of the representations looks different, in the end we always have that the breaking is $O(N_c - k) \to O(N_c - k - 1)$ with transverse slice $c_{N_f-k}$. The chain of Higgsings only stops

after $N_c$ steps, when we have $O(1) \rightarrow \{1\}$ with a $c_{N_f-(N_c-1)}$ transverse slice. The resulting Higgs branch Hasse diagram is a line, and is depicted in the bottom left of Figure 1.

As a byproduct of this analysis, we can also obtain the Hasse diagram for a theory with $SO(N_c)$ gauge group and $N_f$ fundamentals.[4] The process is completely analogous to the one above, except there are no $\mathbb{Z}_2$ representations making any appearance. At each step the possible Higgsing is $SO(N_c - k) \rightarrow SO(N_c - k - 1)$ with transverse slice $c_{N_f-k}$. The only difference comes after $N_c - 2$ steps, when the theory we have left is $SO(2)$ with $N_f - N_c + 2$ fundamentals. In the $O$ case, there was still one possible nontrivial subgroup and Higgsing $O(2) \rightarrow O(1) = \mathbb{Z}_2$. On the other hand, now we have $SO(2) = U(1)$, which has no nontrivial subgroups to be Higgsed to. Therefore the only Higgsing is $U(1) \rightarrow \{1\}$, with transverse slice $a_{2N_f-2N_c+3}$. We show this Hasse diagram in the bottom right of Figure 1. The relation between the Hasse diagrams for the O and SO theories is reminiscent of the relation between U and SU [34, 35], as made clear on the figure.

## 3.2 Hasse diagram for $\widetilde{SU}(N)_I$

We now proceed to compute the Higgs branch Hasse diagrams for theories with $\widetilde{SU}(N_c)_I$ gauge group, and matter content consisting of $N_f$ fields in the $(F \oplus \overline{F})$ representation and $N_\epsilon$ fields in the $\epsilon$ representation. We do this by considering all the possible Higgsing patterns, using the branching rules summarised in Table 3. We consider only the case where $N_f$ is large enough so that we can have complete Higgsing.

Let's begin with the simple example of $\widetilde{SU}(4)_I$ with 4 fundamentals as an appetizer. This representation is real, and so the theory has $Sp(4)$ global symmetry. Computing the Higgsing to $\widetilde{SU}(3)_I$, we find,

$$4(F \oplus \overline{F})_{\widetilde{SU}(4)_I} - \text{Adj}_{\widetilde{SU}(4)_I} \rightarrow 4\left[(F \oplus \overline{F})_{\widetilde{SU}(3)_I} \oplus \epsilon \oplus \mathbf{1}\right] \tag{3.11}$$
$$- \left[\text{Adj}_{\widetilde{SU}(3)_I} \oplus (F \oplus \overline{F})_{\widetilde{SU}(3)_I} \oplus \epsilon\right]$$
$$= 3(F \oplus \overline{F})_{\widetilde{SU}(3)_I} \oplus 3\,\epsilon - \text{Adj}_{\widetilde{SU}(3)_I} \underbrace{\oplus 4 \cdot \mathbf{1}}_{c_4 \text{ slice}}, \tag{3.12}$$

and we see that the remaining theory is an $\widetilde{SU}(3)_I$ gauge theory with 3 fundamentals and 3 fields in the $\epsilon$. Since both the $(F \oplus \overline{F})_{\widetilde{SU}(3)_I}$ and the $\epsilon$ are real representations, this theory has a $Sp(3) \times Sp(3)$ global symmetry. On the other hand, we observe that the transverse slice at the bottom of the Hasse diagram is $c_4$.

We are now presented with two options regarding how to continue the chain of Higgsings. The gauge fields can either eat the fields in the $\epsilon$, resulting on the breaking $\widetilde{SU}(3)_I \rightarrow SU(3)$, or fields in the $(F \oplus \overline{F})_{\widetilde{SU}(3)_I}$, resulting on the breaking $\widetilde{SU}(3)_I \rightarrow SU(2) \times \mathbb{Z}_2$. In the first case, we find

$$3(F \oplus \overline{F})_{\widetilde{SU}(3)_I} \oplus 3\,\epsilon - \text{Adj}_{\widetilde{SU}(3)_I} \rightarrow 6\,F_{SU(3)} - \text{Adj}_{SU(3)} \oplus \underbrace{3 \cdot \mathbf{1}}_{c_3 \text{ slice}}. \tag{3.13}$$

---

[4]We are indebted to Amihay Hanany for drawing our attention to the Hasse diagram of that theory.

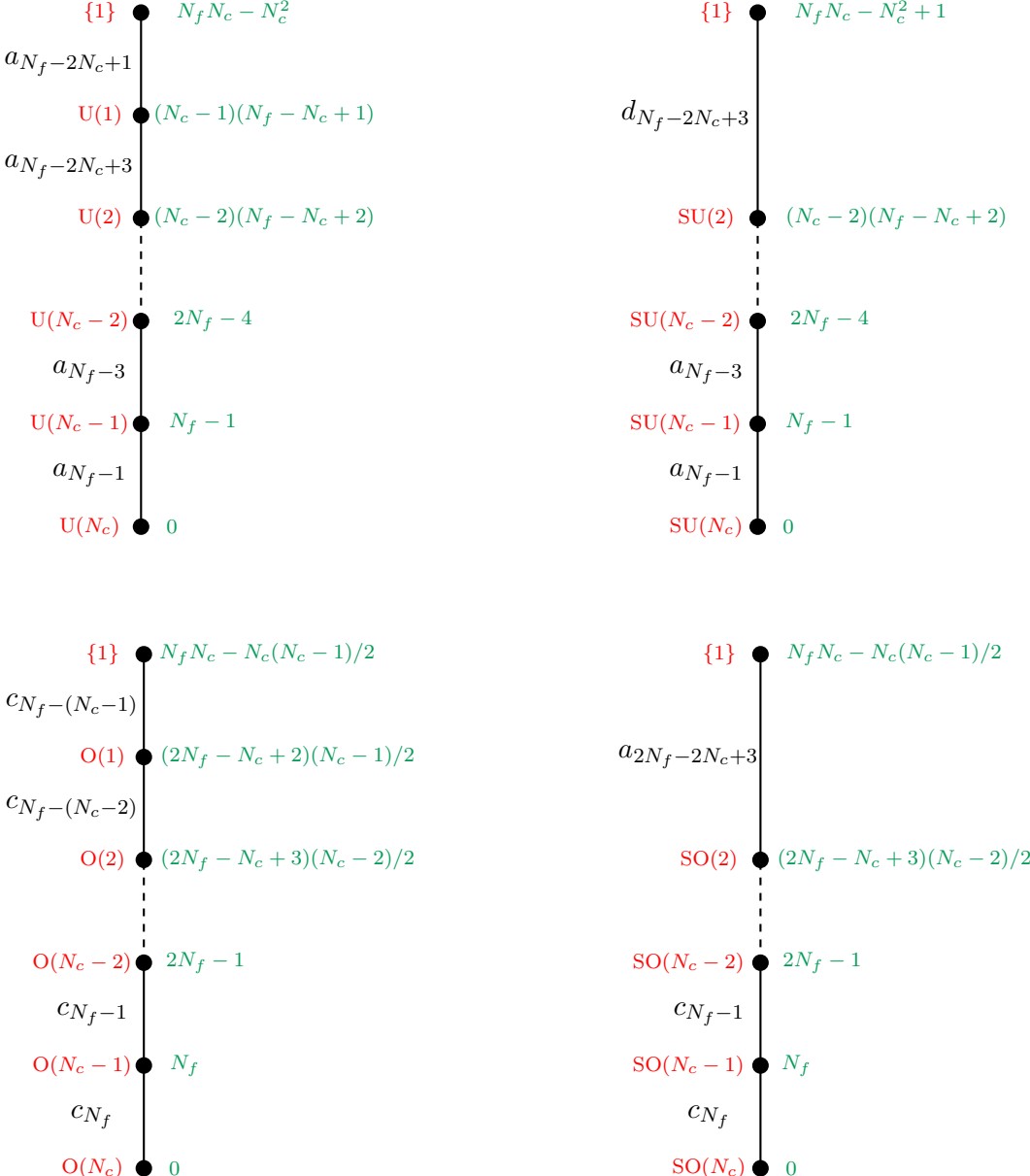

**Figure 1**. Comparison between the Higgs branch Hasse diagram for theories with $\mathrm{U}(N_c)$ (top left) and $\mathrm{SU}(N_c)$ (top right) and for theories with $\mathrm{O}(N_c)$ (bottom left) and $\mathrm{SO}(N_c)$ (bottom right) gauge groups, with the number of fundamental flavours $N_f$ satisfying $N_f \geq N_c$. The green numbers are the quaternionic dimensions of the leaves, and the red groups are the residual gauge groups.

From this point onward, we have the already known Higgsing pattern and Hasse diagram of the SU groups. Let's then consider the second case,

$$3(F \oplus \overline{F})_{\widetilde{SU}(3)_I} \oplus 3\,\epsilon - \text{Adj}_{\widetilde{SU}(3)_I} \to 3\left[ (F \oplus \overline{F})_{\widetilde{SU}(2)_I} \oplus \epsilon \oplus \mathbf{1} \right] \tag{3.14}$$
$$+ 3\,\epsilon - \left[ \text{Adj}_{\widetilde{SU}(2)_I} \oplus (F \oplus \overline{F})_{\widetilde{SU}(2)_I} \oplus \epsilon \right].$$

Here we are making an abuse of notation: since the principal extension of SU(2) is trivial, we have $\widetilde{SU}(2) = SU(2) \times \mathbb{Z}_2$, and the $(F \oplus \overline{F})$ representation is in fact reducible and equal to $2 \cdot F_{SU(2)}$. Using this,

$$(3.14) = 4 \cdot F_{SU(2)} \oplus 5 \cdot \epsilon - \text{Adj}_{SU(2)} \oplus \underbrace{3 \cdot \mathbf{1}}_{c_3 \text{ slice}}. \tag{3.15}$$

We conclude that the theory after this last Higgsing splits into two decoupled theories, one consisting of SU(2) gauge group with 4 flavours and the other of a $\mathbb{Z}_2$ gauge group with 5 fields in the $\epsilon$. The overall global symmetry is $SO(8) \times Sp(5)$.

At this point, the possible Higgsings are trivial, since there are no more nontrivial subgroups. We can either Higgs $SU(2) \to \mathbf{1}$ with the fundamental flavours, leaving the $\mathbb{Z}_2 + 5\epsilon$ alone (this produces a $d_4$ slice), or Higgs $\mathbb{Z}_2 \to \mathbf{1}$ with the $\epsilon$ fields (this produces a $c_5$ slice). Note that the $SU(2) + 4F$ remaining in this transition can also be reached from the Higgsing of $SU(3) + 6F$ that we obtained in the previous steps. All in all, the Hasse diagram for the Higgs branch of this theory is depicted in Figure 2.

Generalizing to an arbitrary (large enough) number of fundamentals and fields in the $\epsilon$ is now straightforward. As before, we begin by writing down the hyperKähler quotient

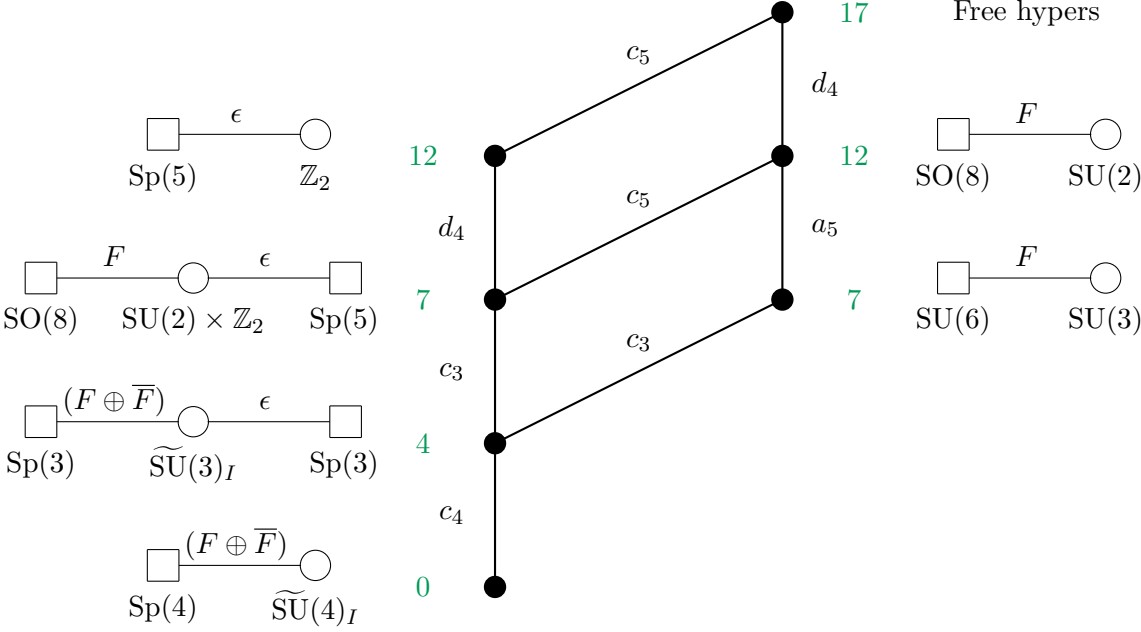

**Figure 2**. Higgs branch Hasse diagram of $\widetilde{SU}(4)_I + 4\,(F \oplus \overline{F})$. Next to each symplectic leaf, we write its dimension (in green) and the quiver of the effective theory.

for the Higgs branch of $\widetilde{\mathrm{SU}}(N_c)_I + N_f(F \oplus \overline{F}) + N_\epsilon \epsilon$,

$$N_f(F \oplus \overline{F})_{\widetilde{\mathrm{SU}}(N_c)} \oplus N_\epsilon \epsilon - \mathrm{Adj}_{\widetilde{\mathrm{SU}}(N_c)_I} . \tag{3.16}$$

This theory has $\mathrm{Sp}(N_f) \times \mathrm{Sp}(N_\epsilon)$ global symmetry. Similarly to the intermediate step of the previous example, there are two possible Higgsings, with the $\epsilon$'s or with the fundamentals. Applying the branching rules of Table 3 in either case results in

$$\widetilde{\mathrm{SU}}(N_c)_I \to \mathrm{SU}(N_c): \quad (3.16) \to N_f F_{\mathrm{SU}(N_c)} \oplus N_f \overline{F}_{\mathrm{SU}(N_c)} \tag{3.17}$$
$$- \mathrm{Adj}_{\mathrm{SU}(N_c)} \underbrace{\oplus N_\epsilon \cdot \mathbf{1}}_{c_{N_\epsilon} \text{ slice}}$$

$$\widetilde{\mathrm{SU}}(N_c)_I \to \widetilde{\mathrm{SU}}(N_c-1)_I: \quad (3.16) \to (N_f-1)(F \oplus \overline{F})_{\widetilde{\mathrm{SU}}(N_c-1)_I} \tag{3.18}$$
$$\oplus (N_\epsilon + N_f - 1)\epsilon - \mathrm{Adj}_{\widetilde{\mathrm{SU}}(N_c-1)} \underbrace{\oplus N_f \cdot \mathbf{1}}_{c_{N_f} \text{ slice}}$$

The former leads to an effective theory with $\mathrm{SU}(N_c)$ gauge group and $\mathrm{SU}(2N_f)$ global symmetry, whose Hasse diagram is already known. This is the right part of the Hasse diagram of Figure 3. The latter leads to a theory with $\widetilde{\mathrm{SU}}(N_c - 1)_I$ gauge group, $N_f - 1$ fields in the $(F \oplus \overline{F})$, and $N_\epsilon + N_f - 1$ fields in the $\epsilon$; thus a $\mathrm{Sp}(N_f - 1) \times \mathrm{Sp}(N_\epsilon + N_f - 1)$ global symmetry. In order to continue the computation of the Hasse diagram, we are once again presented with two possibilities: Higgsing with the $\epsilon$ fields –this produces a $c_{N_\epsilon + N_f - 1}$ slice that merges with the right part of the Hasse diagram corresponding to the connected gauge groups– or with the fundamentals –this produces a $c_{N_f - 1}$ slice that continues on the left side of the Hasse diagram corresponding to the disconnected gauge groups with both fundamental and $\epsilon$ matter fields–.

The Hasse diagram of Figure 3 is the result of iterating this procedure $N_c - 2$ times, until we reach $\widetilde{\mathrm{SU}}(2)_I = \mathrm{SU}(2) \times \mathbb{Z}_2$. At this point, as happened with the previous example, the theory will decouple into an $\mathrm{SU}(2)$ gauge theory with $N_f - N_c + 2$ fundamentals, and a $\mathbb{Z}_2$ gauge theory with $N_\epsilon + (N_c - 2)(2N_f - N_c + 1)/2$ $\epsilon$'s. We can Higgs each of these two gauge groups separetly, resulting in the "rectangle" at the top of the Hasse diagram.

## 3.3 Hasse diagram for $\widetilde{\mathrm{SU}}(N)_{II}$

The computation of the Hasse diagram of the Higgs branch for theories with $\mathrm{SU}(2N_c)_{II}$ gauge group is very similar to the one we just described in detail for the type $I$ case. There are only a few key differences that we need to take into account. The first is that $\widetilde{\mathrm{SU}}(2N-1)_I$ is not a subgroup of $\widetilde{\mathrm{SU}}(2N)_{II}$. This implies that the smallest step we can take in the chain of Higgsings is $\widetilde{\mathrm{SU}}(2N)_{II} \to \widetilde{\mathrm{SU}}(2N-2)_{II}$. The second is that the fundamental representation of the type II groups is pseudo-real rather than real, and therefore these fields will give rise to an SO global symmetry.

As in the type I case, before considering the fully general case, we begin by looking at a concrete example, $\widetilde{\mathrm{SU}}(6)_{II}$ with 6 fields in the $(F \oplus \overline{F})$; the resulting Hasse diagram is depicted in Figure 4. The procedure is the same as before: we begin by writing down

$$6(F \oplus \overline{F})_{\widetilde{\mathrm{SU}}(6)_{II}} - \mathrm{Adj}_{\widetilde{\mathrm{SU}}(6)_{II}} , \tag{3.19}$$

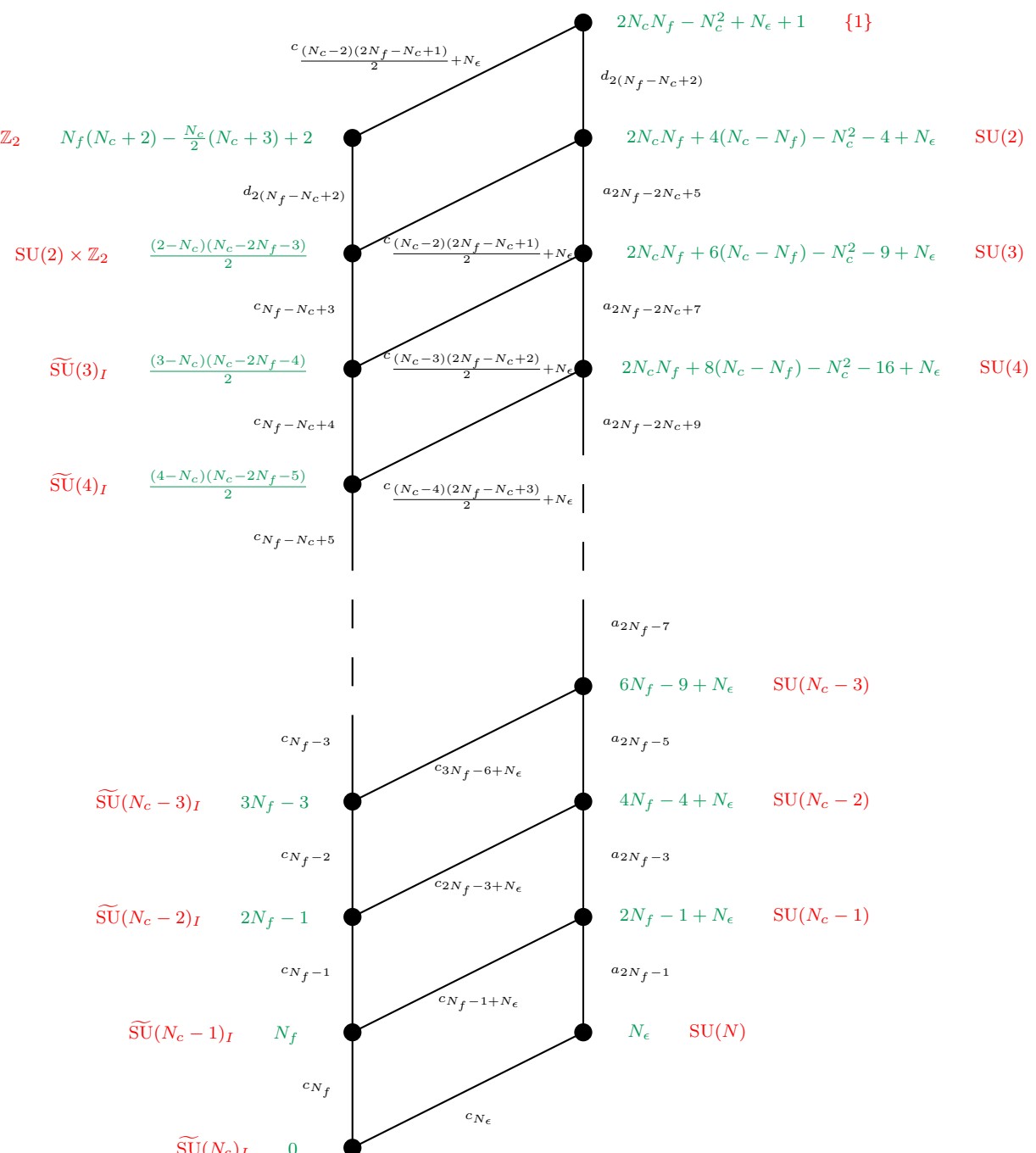

**Figure 3**. Higgs branch Hasse diagram of $\widetilde{\mathrm{SU}}(N_c)_I + N_f\,(F \oplus \overline{F}) + N_\epsilon\,\epsilon$ for $N_f \geq N_c$. The green numbers are the quaternionic dimensions of the leaves, and the red groups are the residual gauge groups.

and apply the branching rules of Table 3 under the breaking $\widetilde{\mathrm{SU}}(6)_{II} \to \widetilde{\mathrm{SU}}(4)_{II}$. After some cancellations, this results in

$$4(F \oplus F)_{\widetilde{\mathrm{SU}}(4)_{II}} \oplus 11\epsilon - \mathrm{Adj}_{\widetilde{\mathrm{SU}}(4)_{II}} \underbrace{\oplus (12 - 3) \cdot \mathbf{1}}_{d_6 \text{ slice}} . \tag{3.20}$$

We see that the remaining effective theory has $\mathrm{SO}(8) \times \mathrm{Sp}(11)$ global symmetry, and the transverse slice, according to Table 4, is the minimal nilpotent orbit of $\mathfrak{so}(12)$. Again we are at a stage where we can proceed with the chain of Higgsings in two ways: either Higgs with the $\epsilon$ fields –this results in a $c_{11}$ slice that goes to the right side of the Hasse diagram corresponding to the SU groups– or with the fundamentals –this results in a $d_4$ slice that continues in the left of the Hasse diagram corresponding to the disconnected groups–. Note that since in the disconnected side of the diagram the rank of the gauge group jumps by two, we will have extra symplectic leaves in the right side of the Hasse diagram. In our example, the extra leaf is the one of dimension 27, with gauge group SU(3) and global symmetry SU(6); meanwhile on the left side we jump directly from $\widetilde{\mathrm{SU}}(4)_{II} \to \mathrm{SU}(2) \times \mathbb{Z}_2$. As in the type I case, the disconnected version of the SU(2) group is simply a direct product, which means that at the top of the Hasse diagram we have a rectangle where the sides are the transverse slices $c_{18}$ –corresponding to Higgsing the $\mathbb{Z}_2$ with the $\epsilon$ fields– and $d_4$ –corresponding to Higgsing the SU(2) with the fundamentals–.

With this in mind, generalizing to a $\widetilde{\mathrm{SU}}(2N_c)_{II} + N_f(F \oplus \overline{F}) + N_\epsilon \epsilon$ (with $N_f$ large enough) requires no extra thinking. We only need to repeat the computation above a few times to obtain the result in Figure 5.

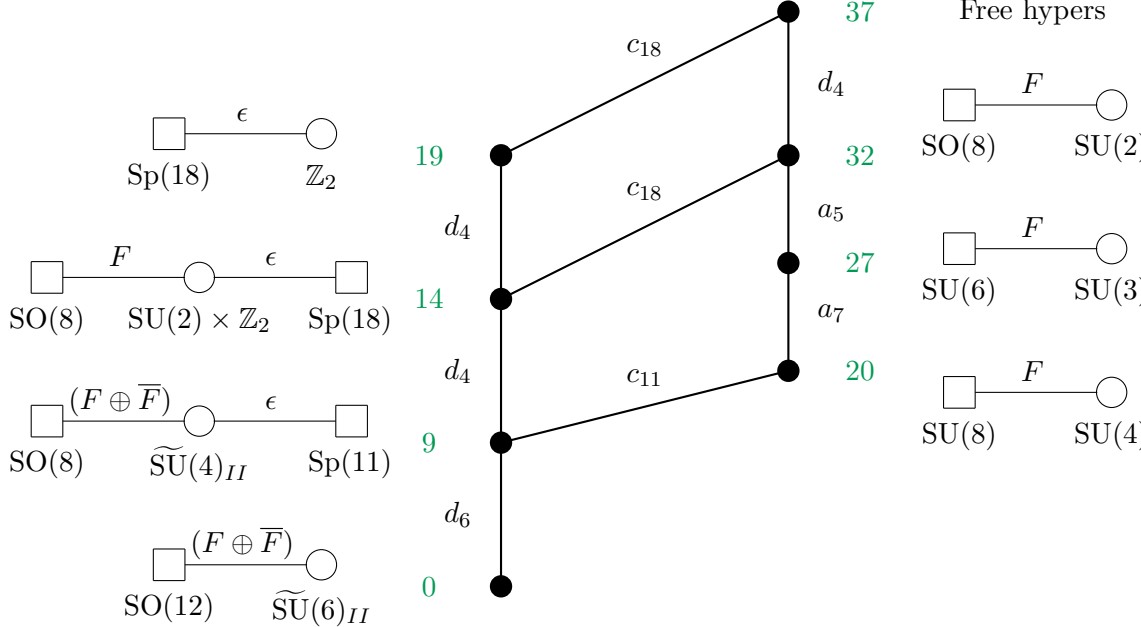

**Figure 4**. Hasse diagram of $\widetilde{\mathrm{SU}}(6)_{II} + 6(F \oplus \overline{F})$. Next to each symplectic leaf, we write its quaternionic dimension (in green) and the quiver of the effective theory corresponding to the transverse slice from that leaf to the top leaf.

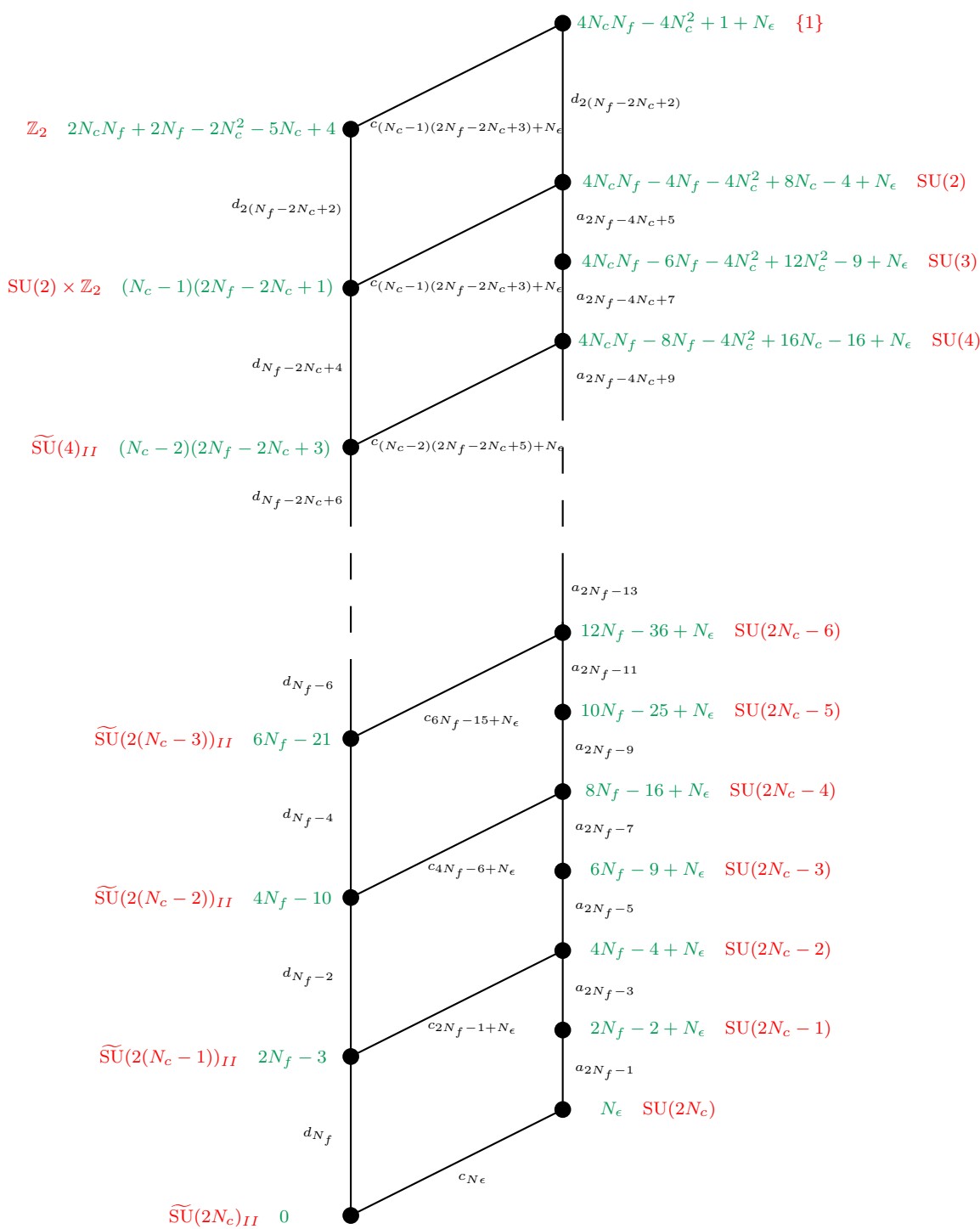

**Figure 5**. Higgs branch Hasse diagram of $\widetilde{\text{SU}}(2N_c)_{II} + N_f(F \oplus \overline{F}) + N_\epsilon \epsilon$ for $N_f \geq 2N_c$. The green numbers are the quaternionic dimensions of the leaves, and the red groups are the residual gauge groups.

## 4 Magnetic quivers

In [1, 28] we began the study of the Higgs branch of the 4d $\mathcal{N} = 2$ discrete gauged SQCD-like theories of type I and II. In this section we attempt to find a magnetic quiver, i.e a 3d $\mathcal{N} = 4$ theory whose Coulomb branch is equal to the Higgs branch of the $\widetilde{\mathrm{SU}}(N)$ gauge theory. We conjecture that the Higgs branch of the 4d $\mathcal{N} = 2$ $\widetilde{\mathrm{SU}}_I(N)$ gauge theory with $N_f(F \oplus \overline{F})$ hypermultiplets is the *wreathed quivers* drawn in Figure 8. We check our conjecture performing the computation of the corresponding Coulomb branch Hilbert series on a selection of examples. Our main tool will be the monopole formula that was initially introduced in [31]. The generalization of this formula in the context of wreathed quivers was performed in [39].

We start with a short review of the monopole formula of [39] before applying it to the theories of interest. We work out with full details the $\widetilde{\mathrm{SU}}(3)_I$ case with $N_f = 3$ while we just report the result for $\widetilde{\mathrm{SU}}_I(N)$ with $N > 3$. The type II theories are discussed in Section 4.4.

### 4.1 Review of the monopole formula

We consider a 3d $\mathcal{N} = 4$ simply laced quiver with unitary nodes and only bifundamental hypermultiplets, and a finite subgroup $\Gamma$ of the automorphisms of that quiver. We call $V$ the set of vertices of the quiver; to each vertex $v \in V$ is associated a unitary gauge group $\mathrm{U}(n_v)$. We call $E$ the set of (unoriented) edges $e = \{v, v'\}$ of the quiver, which correspond to hypermultiplets in bifundamental representations connecting the gauge nodes $v$ and $v'$. The gauge group of the initial quiver is $G = \prod_{v \in V} \mathrm{U}(n_v)$. Wreathing the quiver by $\Gamma$ means promoting the gauge group to the wreath product $G \wr \Gamma$ – see [39] for a reminder of the definition of this operation.[5]

Following [31, 39] the (unrefined) Coulomb branch Hilbert series associated to $\Gamma$ takes the form

$$\mathrm{HS}_\Gamma(t) = \frac{1}{|W_\Gamma|} \sum_{m \in \mathbb{Z}^r} \sum_{\gamma \in W_\Gamma(m)} \frac{t^{2\Delta(m)}}{\det(1 - t^2 \gamma)} \ , \tag{4.1}$$

where $W_\Gamma := W \rtimes \Gamma \subseteq S_{r+1}$ is given by the extension of $W = \prod_{v \in V} S_{n_v}$ by the symmetry $\Gamma$ of the quiver. Here $r = -1 + \sum_{v \in V} n_v$ denotes the total rank of the quiver gauge theory that we are considering, while $m$ denotes the magnetic charge that takes value in the lattice $\mathbb{Z}^r$. For any $m \in \mathbb{Z}^r$ we call $W_\Gamma(m) = \{w \in W_\Gamma \mid w \cdot m = m\}$. Finally $\Delta(m)$ denotes the conformal dimension, defined by

$$2\Delta(m) = \sum_{\{v,v'\} \in E} \sum_{i=1}^{n_v} \sum_{j=1}^{n'_v} |m_{v,i} - m_{v',j}| - \sum_{v \in V} \sum_{i=1}^{n_v} \sum_{j=1}^{n_v} |m_{v,i} - m_{v,j}| . \tag{4.2}$$

When $\Gamma = \{1\}$ is the trivial group, (4.1) reproduces the standard monopole formula of [31]. Henceforth we consider quivers which possess a $\mathbb{Z}_2$ automorphism, and we set $\Gamma = \mathbb{Z}_2$.

---

[5]It should be noted that in the particular case of a symmetry permuting a bouquet of U(1) gauge nodes, the wreathing operation coincides with the discrete gauging of [56–58].

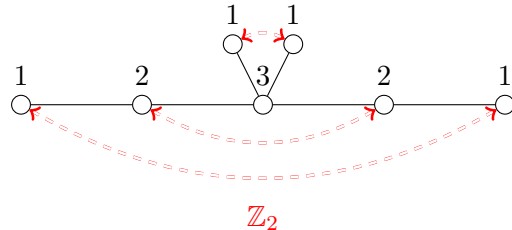

**Figure 6**. Magnetic quiver for SQCD with gauge group $\widetilde{\mathrm{SU}}(3)_I$ and 3 $(F \oplus \overline{F})$. The red dashed lines show the $\mathbb{Z}_2$ action on the legs of the quiver.

Formula (4.1) can be more efficiently evaluated after exploiting the Weyl group symmetry. We introduce the *Casimir factors* $P_{W_\Gamma}$ [6]

$$P_{W_\Gamma}(t;m) = \frac{1}{|W_\Gamma|} \sum_{\gamma \in W_\Gamma(m)} \frac{1}{\det(1 - t^2\gamma)} \ . \tag{4.3}$$

This way the formula (4.1) can be recast in the following form

$$\mathrm{HS}_\Gamma(t) = \sum_{m \in \mathrm{Weyl}(G \wr \Gamma) \cap \mathbb{Z}^r} P_{W_\Gamma}(t;m) t^{2\Delta(m)} \ , \tag{4.4}$$

where $G = \prod_{v \in V} \mathrm{U}(n_v)$ is the initial gauge group and the sum is taken over the magnetic weights in the principal chamber $\mathrm{Weyl}(G \rtimes \Gamma)$.

## 4.2 Example: the $\widetilde{\mathrm{SU}}(3)_I$ case

We start from the magnetic quiver for the Higgs branch of 4d $\mathcal{N} = 2$ SQCD with gauge group SU(3) and $N_f = 3$ flavours. The $\Gamma = \mathbb{Z}_2$ is implemented with a wreathing on the legs of the quiver as schematically shown in Figure 6. Note that the quiver has a full $\mathbb{Z}_2 \times \mathbb{Z}_2$ automorphism group, each factor exchanging two identical legs; we just wreath with the diagonal subgroup $\Gamma$. This is justified by the generalization to $N_f > N$, see Figure 8.

In order to check this conjecture we compute the Coulomb branch Hilbert series using formula (4.1). We believe it is useful to provide the full details of the computation for that example as this is the first time (4.1) is evaluated on a non-trivial wreathed quiver.

- To each gauge node of the quiver we associate the magnetic weights as follows:

$$
\begin{array}{c}
\overset{a}{\bigcirc} \quad \overset{b}{\bigcirc} \\
\bigcirc \!\!-\!\! \bigcirc \!\!-\!\! \underset{g_1, g_2, g_3}{\bigcirc} \!\!-\!\! \bigcirc \!\!-\!\! \bigcirc \\
\underset{c}{} \quad \underset{d_1, d_2}{} \qquad\quad \underset{f_1, f_2}{} \quad \underset{e}{}
\end{array}
\tag{4.5}
$$

The total rank $r$ of the gauge group is $11 - 1 = 10$, and the sum over the magnetic charges is over elements of the form

$$m \cong (a, b, c, d_1, d_2, e, f_1, f_2, g_1, g_2, g_3 = 0) \in \mathbb{Z}^{r+1} \tag{4.6}$$

---

[6]Note that for $W_\Gamma = S_N$ this definition coincides with the definition of the Casimir factors $P_U$ associated to unitary gauge groups, that were introduced in [31].

so this is indeed a sum over $\mathbb{Z}^r$ (see Section 2.4.3 of [59] for detailed explanation about the choice of lattice).

- The Weyl group $W$ is the product of the Weyl groups of the simple gauge groups, namely

$$W = S_1 \times S_1 \times S_1 \times S_2 \times S_1 \times S_2 \times S_3 \subset S_{11} \qquad (4.7)$$

- The wreathing group is $\Gamma = \mathbb{Z}_2$. It is generated by the permutations that exchange simultaneously $a \leftrightarrow b$, $c \leftrightarrow e$ and $d_i \leftrightarrow f_i$ $(i = 1, 2)$. Then the group $W_\Gamma = W \rtimes \Gamma \subset S_{11}$ has order $|W_\Gamma| = 48$.

- The expression (4.2) gives the following conformal dimension for the case at hand

$$2\Delta(m) = \sum_{i=1}^{3} (|a - g_i| + |b - g_i|) + \sum_{i=1}^{2} (|c - d_i| + |e - f_i|) + \sum_{i=1}^{2}\sum_{j=1}^{3} (|d_i - g_j| + |f_i - g_j|)$$

$$- \sum_{i,j=1}^{2} (|d_i - d_j| + |f_i - f_j|) - \sum_{i,j=1}^{3} |g_i - g_j| . \qquad (4.8)$$

We now work out formula (4.4), splitting it into six contributions, one for each generalized wall of the Weyl chamber.

- The interior of the chamber is defined by the inequality $a < b$. In that case, for any $m$ satisfying this inequality, $W_\Gamma(m) = W(m)$ so the Casimir factors correspond to those of $W$, and we get the contribution

$$H_1(t) = \frac{(1 - t^2)}{(1 - t^2)^4} \sum_{a<b} \sum_{c} \sum_{d_1 \le d_2} \sum_{e} \sum_{f_1 \le f_2} \sum_{g_1 \le g_2 \le 0} P_U(d)P_U(f)P_U(g)t^{2\Delta(a,b,c,d_1,d_2,e,f_1,f_2,g_1,g_2,0)} .$$
$$(4.9)$$

Note that we factored out the Casimir terms for the four U(1) nodes, giving $(1 - t^2)^{-4}$ in the denominator, and we include a $(1 - t^2)$ in the numerator to account for the decoupled U(1). In (4.9) and all similar equations below, all sums run over the integers $\mathbb{Z}$.

- Then we go on the wall of the chamber defined by $a = b$. Now to avoid over counting we have to be in the interior of that wall, which we define by the inequality $c < e$. In that case clearly $W_\Gamma(m) = W(m)$, so the contribution is

$$H_2(t) = \frac{(1 - t^2)}{(1 - t^2)^4} \sum_{a} \sum_{c<e} \sum_{d_1 \le d_2} \sum_{f_1 \le f_2} \sum_{g_1 \le g_2 \le 0} P_U(d)P_U(f)P_U(g)t^{2\Delta(a,a,c,d_1,d_2,e,f_1,f_2,g_1,g_2,0)} .$$
$$(4.10)$$

- The third (respectively the fourth) contributions are defined by $a = b$, $c = e$ and $d_2 < f_2$ (respectively $d_2 = f_2$ and $d_1 < f_1$). This uses a lexicographic order to find a fundamental chamber relative to the fugacities of the non-abelian groups U(2). Again

these constraints guarantee that $(a, b, c, d_1, d_2, e, f_1, f_2) \neq (b, a, e, f_1, f_2, c, d_1, d_2)$ so $W_\Gamma(m) = W(m)$ and the contributions are

$$\mathrm{H}_3(t) = \frac{(1-t^2)}{(1-t^2)^4} \sum_a \sum_c \sum_{f_1 \leq f_2} \sum_{d_1 \leq d_2 < f_2} \sum_{g_1 \leq g_2 \leq 0} P_\mathrm{U}(d) P_\mathrm{U}(f) P_\mathrm{U}(g) t^{2\Delta(a,a,c,d_1,d_2,c,f_1,f_2,g_1,g_2,0)} ,$$

(4.11)

$$\mathrm{H}_4(t) = \frac{(1-t^2)}{(1-t^2)^4} \sum_a \sum_c \sum_{f_1 \leq f_2} \sum_{d_1 < f_1} \sum_{g_1 \leq g_2 \leq 0} P_\mathrm{U}(d) P_\mathrm{U}(f) P_\mathrm{U}(g) t^{2\Delta(a,a,c,d_1,f_2,c,f_1,f_2,g_1,g_2,0)} .$$

(4.12)

- We now reach the regions where $(a, b, c, d_1, d_2, e, f_1, f_2) = (b, a, e, f_1, f_2, c, d_1, d_2)$. In that case we can no longer use the standard Casimir factors $P_\mathrm{U}$ for the U(2) nodes. Consider first the fifth region, defined by

$$a = b \qquad c = e \qquad d_1 = f_1 \qquad d_2 = f_2 \qquad f_1 < f_2 . \tag{4.13}$$

As the factor $S_3$ in $W$ is unaffected, we keep the $P_\mathrm{U}$ Casimir term for it. Let us denote $W' = W/S_3 = S_1 \times S_1 \times S_1 \times S_2 \times S_1 \times S_2 \subset S_8$. For a weight $m$ satisfying (4.13), $W'_\Gamma(m)$ does not depend on $m$, so we can factor out from (4.1) a prefactor

$$\frac{1}{|W'_\Gamma|} \sum_{\gamma \in W_\Gamma(m)} \frac{1}{\det(1 - t^2 \gamma)} = \frac{1 + 6t^4 + t^8}{(1 - t^2)^8 (1 + t^2)^4} . \tag{4.14}$$

Therefore the fifth contribution is

$$\mathrm{H}_5(t) = (1-t^2) \frac{1 + 6t^4 + t^8}{(1 - t^2)^8 (1 + t^2)^4} \sum_a \sum_c \sum_{f_1 < f_2} \sum_{g_1 \leq g_2 \leq 0} P_\mathrm{U}(g) t^{2\Delta(a,a,c,f_1,f_2,c,f_1,f_2,g_1,g_2,0)} .$$

(4.15)

In that expression the four U(1) gauge nodes Casimirs are accounted for in (4.14).

- Finally the last region is defined by

$$a = b \qquad c = e \qquad d_1 = f_1 \qquad d_2 = f_2 \qquad f_1 = f_2 . \tag{4.16}$$

and for such an $m$ we get

$$\frac{1}{|W'_\Gamma|} \sum_{\gamma \in W_\Gamma(m)} \frac{1}{\det(1 - t^2 \gamma)} = \frac{1 - t^2 + 4t^4 - t^6 + t^8}{(1 - t^2)^8 (1 + t^2)^4 (1 + t^4)} . \tag{4.17}$$

This gives the contribution

$$\mathrm{H}_6(t) = (1-t^2) \frac{1 - t^2 + 4t^4 - t^6 + t^8}{(1 - t^2)^8 (1 + t^2)^4 (1 + t^4)} \sum_a \sum_c \sum_{f_1} \sum_{g_1 \leq g_2 \leq 0} P_\mathrm{U}(g) t^{2\Delta(a,a,c,f_1,f_1,c,f_1,f_1,g_1,g_2,0)} .$$

(4.18)

The Hilbert series (4.1) for the case at hand is the sum of these six contributions. Evaluating each of them perturbatively, we find

$$\begin{aligned}
\mathrm{HS}(t) = {} & 1 + 21t^2 + 20t^3 + 336t^4 + 560t^5 + 3850t^6 + 7812t^7 + 34643t^8 + 73900t^9 \\
& + 252132t^{10} + 535920t^{11} + 1533810t^{12} + 3177876t^{13} + 8011642t^{14} + 16049712t^{15} \\
& + 36748014t^{16} + O\left(t^{17}\right) .
\end{aligned} \tag{4.19}$$

The Higgs branch Hilbert series for this theory has been evaluated exactly in [28] using the Molien-Weyl integration formula for disconnected groups [60], giving the result

$$\begin{aligned}
& \frac{1}{(1-t)^{20}(1+t)^{16}(1+t^2)^8(1+t+t^2)^{10}} \Big( 1 + 6t + 34t^2 + 144t^3 + 647t^4 + 2588t^5 + \\
& 9663t^6 + 31988t^7 + 97058t^8 + 268350t^9 + 687264t^{10} + 1628374t^{11} + 3598201t^{12} + \\
& 7421198t^{13} + 14364220t^{14} + 26130494t^{15} + 44837750t^{16} + 72656468t^{17} + 111456702t^{18} + \\
& 162010222t^{19} + 223544610t^{20} + 292994926t^{21} + 365233973t^{22} + 433158422t^{23} + \\
& 489154949t^{24} + 526027956t^{25} + 538960928t^{26} + \ \ldots \ + \text{palindrome} \ + \ldots + t^{52} \Big) .
\end{aligned}$$

All computed orders in (4.19) agree with the above expression, giving a strong evidence that the wreathed quiver of Figure 6 can be considered to be a magnetic quiver for the $\widetilde{\mathrm{SU}}(3)_I$ gauge theory with $3\,(F \oplus \overline{F})$. We note that the evaluation of the corresponding refined Hilbert series does not present any conceptual obstruction, although the computational cost quickly becomes prohibitive. We did check the equality of the refined Hilbert series up to order $t^4$.

**Folded quiver and twisted compactification**

In this paragraph, we denote by $\mathcal{C}$ the Coulomb branch of the 3d $\mathcal{N}=4$ theory defined by the wreathed quiver of Figure 6 and by $\mathcal{H}$ the Higgs branch of the 4d $\mathcal{N}=2$ $\widetilde{\mathrm{SU}}(3)_I$ gauge theory with $3(F \oplus \overline{F})$ matter. We now consider the following three claims:

($\alpha$)  The exact Hilbert series of $\mathcal{C}$ and $\mathcal{H}$ are equal.

($\beta$)  The Hasse diagrams of $\mathcal{C}$ and $\mathcal{H}$ as symplectic singularities agree.

($\gamma$)  The symplectic singularities $\mathcal{C}$ and $\mathcal{H}$ are isomorphic.

The logical implications between these statements is $(\gamma) \implies (\beta) \implies (\alpha)$. The computation performed above strongly suggests that $(\alpha)$ holds. Based on that result, and on physical intuition regarding charge conjugation, we conjecture that $(\gamma)$ holds as well. If this is correct, then $(\beta)$ should also be correct, and in combination with the results of Section 3, it means that we have identified the Hasse diagram for $\mathcal{C}$, see the middle column of Table 5. It is interesting to compare this Hasse diagram with the Hasse diagram of a third quiver, namely the non simply laced quiver

$$\tag{4.20}$$

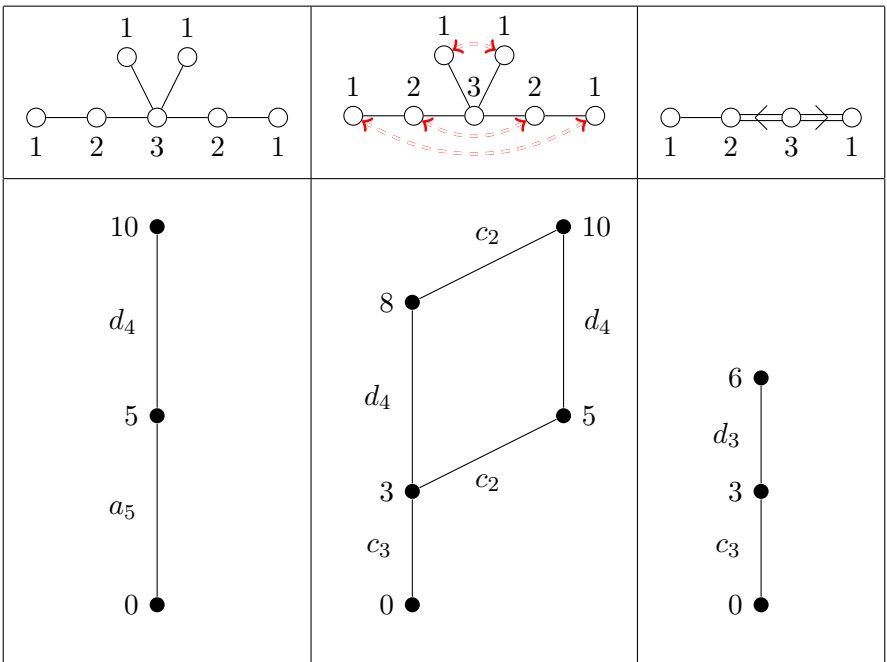

**Table 5**. Comparison of the Hasse diagrams for a quiver with a $\mathbb{Z}_2$ symmetry and the corresponding wreathed and folded quivers.

obtained by folding, presented in the last column of Table 5. The Hasse diagram is obtained from the quiver subtraction algorithm (see [40] for a similar computation).

We note that the quiver (4.20) arises naturally as follows. Consider the following brane web, where vertical lines represent NS5 branes, horizontal lines represent D5 branes and circles represent $(p, q)$-seven-branes with appropriate charges:

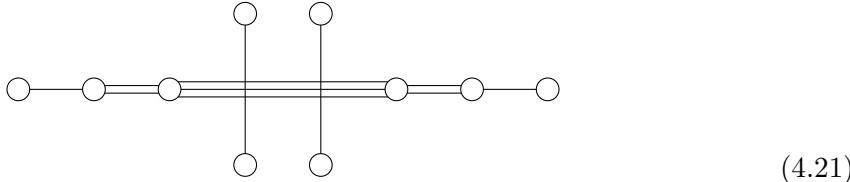

$$(4.21)$$

This represents the 5d $\mathcal{N} = 1$ theory SU(3) with 6 fundamental hypers, with masses set to zero, and finite gauge coupling. This brane web has a $\mathbb{Z}_2 \times \mathbb{Z}_2$ symmetry (the first factor being the reflection with respect to a vertical axis, and the second factor a reflection with respect to a horizontal axis). In particular, the diagonal $\mathbb{Z}_2$, which is a rotation of angle $\pi$ in the plane of the brane web, should correspond to charge conjugation in the SU(3) theory [17]. The magnetic quiver associated to this brane web is

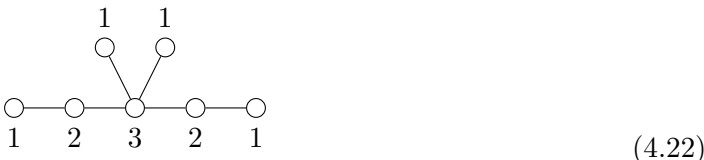

$$(4.22)$$

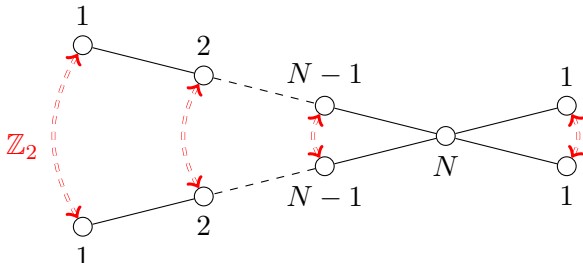

**Figure 7**. Wreathed quiver for SQCD-like theories with gauge group $\widetilde{\mathrm{SU}}(N)_I$ and $N$ flavours. The automorphism group of this quiver is $\mathbb{Z}_2 \times \mathbb{Z}_2$, but we wreath only a $\mathbb{Z}_2$ subgroup, as made clear by the generalization to higher number of flavors in Figure 8.

It has a $\mathrm{SU}(6) \times \mathrm{U}(1)$ global symmetry. We can compactify this 5d theory on a circle with a $\mathbb{Z}_2$ twist, corresponding to charge conjugation, to obtain a $\mathcal{N} = 2$ theory in 4d, following [17]. Then the $\mathrm{SU}(6)$ factor in the global symmetry is broken to $\mathrm{Sp}(3)$, and the $\mathrm{U}(1)$ factor is completely broken. The magnetic quiver, which is derived using the rules of Appendix B of [61], is (4.20).

This construction sheds light on the difference between the wreathed and the folded quivers from the 4d perspective. In the first case, charge conjugation is gauged, which means that inequivalent configurations in the original theory are declared to be equivalent. Mathematically, the operation on the Higgs branch is a quotient, and the dimension is unchanged. In the second case, charge conjugation is involved in twisted compactification: mathematically, the operation on the Higgs branch is a reduction to fixed points of the discrete action, and accordingly the dimension is changed. We conclude this section with an observation of an apparent conflict with a conjecture of [39], which states that the Hasse diagram of a folded quiver should be a subdiagram of the Hasse diagram of any corresponding wreathed quiver. The diagrams of Table 5 contradict this conjecture (which was based on observation of a few examples), and it would be interesting to study this point further.

### 4.3 Type I – general case

Based on the computation performed for the $\widetilde{\mathrm{SU}}(3)_I$ case we can infer the general form of the wreathed magnetic quiver for theories of type I, see Figure 8. When $N_f = N$ this reduces to the quiver of Figure 7, where the $\mathbb{Z}_2$ action is picked from the $\mathbb{Z}_2 \times \mathbb{Z}_2$ automorphism group of the quiver by continuation from the $N_f > N$ case.

For $N \geq 4$ the explicit evaluation of the Coulomb branch Hilbert series for $N_f = N$, associated to the conjectured wreathed quiver, turns out to be computationally quite involved. Due to this obstruction we checked our conjecture only for the case $N = 4$, where the application of the formula (4.1) gives

$$1 + 36t^2 + 1114t^4 + 24717t^6 + 417276t^8 + o(t^8) \ . \tag{4.23}$$

We observe that this expression perfectly matches with the first orders of the expansion of the Higgs branch Hilbert series for SQCD with gauge group $\widetilde{\mathrm{SU}}(4)_I$ and eight flavours,

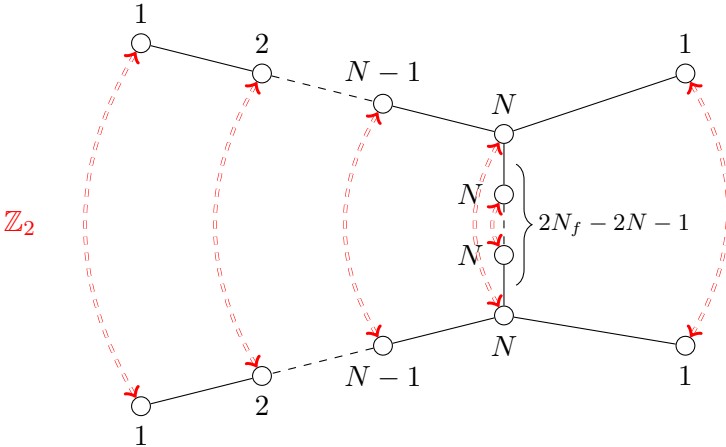

**Figure 8**. Wreathed quiver for SQCD-like theories with gauge group $\widetilde{\text{SU}}(N)_I$ and $N_f > N$ flavours. There is a single $\mathbb{Z}_2$ action which flips the whole quiver about the horizontal axis. The global symmetry of this wreathed quiver is $\text{Sp}(N_f)$.

that was computed in [28].

## 4.4 Type II

In the previous sections, we have provided the magnetic quivers for theories with one of the types of disconnected gauge groups that we have discussed in this article, $\widetilde{\text{SU}}(N)_I$. Currently we have no candidate for a possible magnetic quiver of a theory with $\widetilde{\text{SU}}(N)_{II}$ gauge group. To understand why the type II groups pose a much bigger problem than the type I groups, let's look into the logic that led us to the wreathed quiver in Figure 6.

Two of the main characteristics of the Higgs branch of SQCD-like theories with $\widetilde{\text{SU}}$ gauge groups groups are that, on the one hand, its dimension is the same as for their connected cousins SU, while on the other hand the global symmetry is modified due to the reality properties of the fundamental representation. In particular, for $N_c = 4$ with 4 $F \oplus \overline{F}$ the quaternionic dimension of the Higgs branch is 17, and the global symmetry is SU, Sp or SO in the connected case, type I and type II respectively.

When looking for a magnetic quiver, a natural starting point is the known magnetic quiver for the SU groups, which in the $N_c = 3$ case is depicted in (4.22). This has the correct dimension, but the wrong global symmetry for our purposes. We also have its folded version, the non-simply laced quiver in (4.20); this has the correct global symmetry $\text{Sp}(3)$, but the incorrect dimension. With this in mind, the introduction of wreathed quivers in [39] quickly leads to a potential candidate for the magnetic quiver of $\widetilde{\text{SU}}(3)_I$, since the wreathing construction preserves the dimension, while modifying the global symmetry in the same way as the folding. This candidate is the one in Figure 6, and it turned out to be the correct one.

However, for the type II groups the puzzle is significantly more complicated. Our analysis shows that starting from the magnetic quiver of SU(4), none of the possible ways to wreath a $\mathbb{Z}_2$ gives rise to the expected global symmetry. This has been confirmed by

Hilbert series computations. Thus, as stated above, we have no candidate for the magnetic quiver of $\widetilde{\mathrm{SU}}(N_c)_{II}$. It is of course possible that such a magnetic quiver may be found as a wreathing of a completely different quiver, perhaps including not only unitary nodes; or from an altogether different route.

## 5  Conclusions

In this article we analyzed several aspects of 4d $\mathcal{N} = 2$ theories with disconnected gauge groups. In particular we studied how the global structure of these groups affects the Hasse diagrams for the Higgs branch of supersymmetric gauge theories. The main difference with respect the connected case is that these diagrams are characterized by the presence of bifurcations, physically corresponding to scalar fields transforming in different representations of the gauge group getting a VEV.

Moreover, in the second part of the paper, we moved a further step towards the understanding of the Higgs branch of the 4d $\mathcal{N} = 2$ SQCD like-theories with $\widetilde{\mathrm{SU}}(N)_I$ gauge group providing a candidate for a magnetic quiver that turns out to be a wreathed quiver. Our analysis also suggests that a magnetic quiver for type II theories is not a wreathed quiver of type discussed in [39] or, to the best of our knowledge, any other type 3d $\mathcal{N} = 4$ quiver appearing in the literature. We leave the identification of this quiver for future investigation.

This naturally leads to a wealth of open problems, the most prominent of which being the connection between the two parts of this work, namely the Hasse diagrams and the magnetic quivers. To the best of our knowledge, the algorithms of quiver subtraction leading to Hasse diagrams has not been extended to wreathed quivers. The present work thus offers an infinite family of data points that could serve as a basis to understand how quiver subtraction applies to those quivers. In particular, it should be noted that the Hasse diagram for a wreathed quiver seems not to contain in general the Hasse diagram of the associated folded quiver, as shown in Table 5. This point needs to be investigated further. A brane realization of theories with $\widetilde{\mathrm{SU}}(N)$ gauge groups, possibly along the lines of [62–65] would be an important step forward.

### Acknowledgements

It is a pleasure to thank Julius Grimminger, Amihay Hanany, Rudolph Kalveks, Dominik Miketa, Diego Rodríguez-Gómez and Zhenghao Zhong for helpful comments and discussions. G.A-T would also like to thank the Theoretical Physics Group at Imperial College London for their hospitality during an early stage of this project. The work of AB is supported by STFC grants ST/P000762/1 and ST/T000791/1. AP is supported by "*Borsa di studio I.N.F.N. post-doctoral per fisici teorici.*" G.A-T is supported by the Spanish government scholarship MCIU-19-FPU18/02221. He also acknowledges support from the Principado de Asturias through the grant FC-GRUPIN-IDI/2018/000174.

# A  The groups $\widetilde{\mathrm{SU}}(N)$ and their characters

## A.1  Definition of $\widetilde{\mathrm{SU}}(N)$

**The groups.**    We are interested in semidirect products $\mathrm{SU}(N) \rtimes_\Theta \mathbb{Z}_2$, defined by a group morphism $\Theta : \mathbb{Z}_2 \to \mathrm{Aut}(\mathrm{SU}(N))$. There are essentially two inequivalent choices for $\Theta$, see Table 2 in [28]. For $g \in \mathrm{SU}(N)$, we define $\Theta^I_{+1}(g) = \Theta^{II}_{+1}(g) = g$ and

$$\Theta^I_{-1}(g) = (g^{-1})^T = \overline{g}, \qquad \Theta^{II}_{-1}(g) = -J_N(g^{-1})^T J_N = -J_N \overline{g} J_N \ , \tag{A.1}$$

where the bar denotes complex conjugation and the matrix $J_{2N}$ reads

$$J_{2N} := \begin{pmatrix} 0 & -\mathbb{I}_{N \times N} \\ \mathbb{I}_{N \times N} & 0 \end{pmatrix} \ . \tag{A.2}$$

Moreover we note that $\Theta^{II}_{-1}$ is defined only for $N$ even. When we discuss both cases together, we simply use the letter $\Theta$. Spelling out the definition of the semidirect product, the group $\widetilde{\mathrm{SU}}(N)_{I,II}$ is the Cartesian product $\mathrm{SU}(N) \times \mathbb{Z}_2$ with group law defined by

$$(g, \epsilon) \cdot (g', \epsilon') = (g\Theta_\epsilon(g'), \epsilon\epsilon') \,. \tag{A.3}$$

Explicitly, we can write $\widetilde{\mathrm{SU}}(N)_{I,II}$ as a union of two connected components

$$\widetilde{\mathrm{SU}}(N)_{I,II} = \{(g, 1) \mid g \in \mathrm{SU}(N)\} \cup \{(g, -1) \mid g \in \mathrm{SU}(N)\} \tag{A.4}$$

with the product rules

$$(g, 1) \cdot (g', 1) = (gg', 1) \tag{A.5}$$

$$(g, 1) \cdot (g', -1) = (gg', -1) \tag{A.6}$$

$$(g, -1) \cdot (g', 1) = (g\Theta(g'), -1) \tag{A.7}$$

$$(g, -1) \cdot (g', -1) = (g\Theta(g'), 1) \,. \tag{A.8}$$

From this we also have

$$(g, \epsilon)^{-1} = (\Theta_\epsilon(g^{-1}), \epsilon) \,. \tag{A.9}$$

and

$$(g', \epsilon') \cdot (g, \epsilon) \cdot (g', \epsilon')^{-1} = (g'\Theta_{\epsilon'}(g)\Theta_\epsilon(g')^{-1}, \epsilon) \,. \tag{A.10}$$

**The Lie Algebra.**    The Lie algebra of $\widetilde{\mathrm{SU}}(N)_{I,II}$ is

$$\mathfrak{g} = \{X \in \mathfrak{gl}(N, \mathbb{C}) \mid \mathrm{Tr}(X) = 0 \text{ and } X + X^\dagger = 0\} \,. \tag{A.11}$$

The involutions $\Theta^{I,II}_{-1}$ on $\widetilde{\mathrm{SU}}(N)_{I,II}$ descend to involutions on the Lie algebra defined by

$$\theta^I_{-1}(X) = -X^T, \qquad \theta^{II}_{-1}(X) = J_N X^T J_N \,. \tag{A.12}$$

This is also valid on the complexified Lie algebra, where the condition that $X + X^\dagger = 0$ is dropped. We can rewrite equation (A.10) for $(g, \epsilon) = (1 + X, 1) \equiv 1 + X$ with $X \in \mathfrak{g}$, and get the adjoint representation of $\widetilde{\mathrm{SU}}(N)_{I,II}$:

$$(g, \epsilon) \cdot X \cdot (g, \epsilon)^{-1} = g\theta_\epsilon(X)g^{-1} \,. \tag{A.13}$$

It is useful to compute the trace of $\theta^{I,II}$, and this can be done by expressing it on any basis of $\mathfrak{g}$. We use as a basis $\{(A_{ij})_{1 \leq i < j \leq N}, (B_{ij})_{1 \leq i < j \leq N}, (C_i)_{1 \leq i < N}\}$ with $(A_{ij})_{kl} = \delta_{ik}\delta_{jl} - \delta_{jk}\delta_{il}$, $(B_{ij})_{kl} = i\left(\delta_{ik}\delta_{jl} + \delta_{jk}\delta_{il}\right)$ and $(C_i)_{kl} = i\left(\delta_{ik}\delta_{il} - \delta_{i+1,k}\delta_{i+1,l}\right)$. The matrices $A$ are eigenvectors of $\theta^I$ with eigenvalue $+1$ and the matrices $B$ and $C$ are eigenvectors with eigenvalue $-1$, so

$$\mathrm{Tr}\left(\theta^I\right) = 1 - N. \tag{A.14}$$

For $\theta^{II}$ with $N = 2n$ even, we note that the matrices $A$ and $B$ are permuted (with signs) and the eigenvectors are $A_{i,i+n}$ and $B_{i,i+n}$ with eigenvalue $+1$. Finally there is a contribution $+1$ from $\theta^{II}(C_n) = \sum_{1 \leq i < N} C_i$, so the trace of $\theta^{II}$ is $2n + 1$:

$$\mathrm{Tr}\left(\theta^{II}\right) = 1 + N. \tag{A.15}$$

## A.2 Maximal tori and Cartan Subgroups

Before writing characters for representation of a Lie group $G$, it is necessary to pick a subgroup which is parametrized by a collection of variables $z_i$ (called fugacities, which can assume continuous or discrete range). For connected compact Lie groups, there is an obvious choice, which is a maximal torus $\mathrm{U}(1)^r$ where $r$ is the rank of the group. The situation is much less clear when one considers disconnected groups. For general considerations, we refer the reader to [66, Chapter VII] and [67, Chapter I] for a discussion of the various Cartan subgroups, and to the series of papers by Lusztig starting with [68] for characters of disconnected groups. The case of $\widetilde{\mathrm{SU}}(N)_{I,II}$ is discussed more specifically in [69].

Here we simply give a brief and explicit exposition of the situation in the simplest non trivial case of $\widetilde{\mathrm{SU}}(3)_I$, the generalization to $\widetilde{\mathrm{SU}}(N)_{I,II}$ being straightforward.

Let us define the diagonal and anti-diagonal matrices

$$D(z_1, z_2, z_3) = \begin{pmatrix} z_1 & 0 & 0 \\ 0 & z_2 & 0 \\ 0 & 0 & z_3 \end{pmatrix} \qquad A(z_1, z_2, z_3) = - \begin{pmatrix} 0 & 0 & z_1 \\ 0 & z_2 & 0 \\ z_3 & 0 & 0 \end{pmatrix}. \tag{A.16}$$

The minus sign is there to ensure that $\det D(z_1, z_2, z_3) = \det A(z_1, z_2, z_3) = z_1 z_2 z_3$. We have

$$D(z_1, z_2, z_3) \in \mathrm{SU}(3) \iff A(z_1, z_2, z_3) \in \mathrm{SU}(3) \iff |z_1| = |z_2| = |z_3| = z_1 z_2 z_3 = 1 \tag{A.17}$$

Obviously we have a group morphism $T = \mathrm{U}(1)^2 \to \mathrm{SU}(3)$ given by $(z_1, z_2) \mapsto D\left(z_1, \frac{z_2}{z_1}, \frac{1}{z_2}\right)$. $T$ has two interesting properties:

**A.** It is a maximal torus[7] of $\mathrm{SU}(3)$.

**B.** It is a large Cartan subgroup [67] of $\mathrm{SU}(3)$, i.e. it is equal to the set of elements that normalize a certain maximal torus (namely itself) and fixes the fundamental Weyl chamber.

---

[7]A maximal torus is a compact, connected, abelian subgroup.

**C.** Any element in SU(3) is conjugate to at least one element of $T$.

We want to see how this can be extended to $\widetilde{\text{SU}}(3)$. The crucial point is that the three properties **A**, **B** and **C** are not equivalent in the context of disconnected groups.

In SU(3), $T$ is still a maximal torus. The corresponding large Cartan subgroup is the set of elements $g \in \widetilde{\text{SU}}(3)$ such that $g^{-1}Tg = T$ and $g^{-1}Bg = B$ where $B$ is the set of elements of the form $(M, 1)$ with $M$ upper triangular. We find that the large Cartan subgroup is given by

$$T^+ = \{\varphi(z_1, z_2, \epsilon) \mid z_1, z_2 \in \text{U}(1), \quad \epsilon = \pm 1\} , \tag{A.18}$$

where we have defined

$$\varphi(z_1, z_2, \epsilon) = \begin{cases} \left( D\left( z_1, \frac{z_2}{z_1}, \frac{1}{z_2} \right), 1 \right) & \text{if } \epsilon = 1 \\ \left( A\left( z_1, \frac{z_2}{z_1}, \frac{1}{z_2} \right), -1 \right) & \text{if } \epsilon = -1 . \end{cases} \tag{A.19}$$

The product rules give

$$\varphi(z_1, z_2, \epsilon) \cdot \varphi(y_1, y_2, \eta) = \begin{cases} \varphi(z_1 y_1, z_2 y_2, \epsilon\eta) & \text{if } \epsilon = 1 \\ \varphi(z_1 y_2, z_2 y_1, \epsilon\eta) & \text{if } \epsilon = -1 . \end{cases} \tag{A.20}$$

This means that $\varphi$ is an injective group morphism $\text{U}(1)^2 \rtimes \mathbb{Z}_2 \to \widetilde{\text{SU}}(3)$ where the semidirect product $\text{U}(1)^2 \rtimes \mathbb{Z}_2$ is defined by

$$(z_1, z_2, \epsilon) \cdot (y_1, y_2, \eta) = \begin{cases} (z_1 y_1, z_2 y_2, \epsilon\eta) & \text{if } \epsilon = 1 \\ (z_1 y_2, z_2 y_1, \epsilon\eta) & \text{if } \epsilon = -1 , \end{cases} \tag{A.21}$$

so that the semidirect product can be identified with the wreath product $\text{U}(1) \wr S_2$. Clearly, this group is not Abelian, and as a consequence its image $T^+$ by $\varphi$ is not Abelian either.

A natural Abelian subgroup of $\text{U}(1) \wr S_2$ is $T = \text{U}(1)^2$ considered above. This is in fact the *small Cartan subgroup* [67] associated to $T$, defined as the centralizer of $T$, which in the present case is equal to $T$. Clearly this is not relevant for our study of the disconnected component of $\widetilde{\text{SU}}(3)$.

Another natural Abelian subgroup is $\text{U}(1) \times \mathbb{Z}_2$ where the first factor is the diagonal subgroup of $T$. Its image in $\widetilde{\text{SU}}(3)$ is

$$T^0 = \{\varphi(z, z, \epsilon) \mid z \in \text{U}(1), \quad \epsilon = \pm 1\} . \tag{A.22}$$

Property **C** fails here: clearly not every element of $\widetilde{\text{SU}}(3)$ is conjugate to an element of $T^0$. Note however that every element of the disconnected part of $\widetilde{\text{SU}}(3)$ is conjugate to an element of the disconnected part of $T^0$. This property is crucial in establishing a Weyl integration formula over $\widetilde{\text{SU}}(3)$ [60].

Finally, consider the subgroup

$$\mathcal{T} = \{\psi(z_1, z_2, \epsilon) \mid z_1, z_2 \in \text{U}(1), \quad \epsilon = \pm 1\} , \tag{A.23}$$

| Group | $F \oplus \overline{F}$ character |
|---|---|
| $T = T^-$ | $z_1 + \frac{z_2}{z_1} + \frac{1}{z_2} + z_2 + \frac{z_1}{z_2} + \frac{1}{z_1}$ |
| $T^+$ | $\left(\frac{1+\epsilon}{2}\right)\left(z_1 + \frac{z_2}{z_1} + \frac{1}{z_2} + z_2 + \frac{z_1}{z_2} + \frac{1}{z_1}\right)$ |
| $T^0$ | $(1 + \epsilon)\left(z + 1 + \frac{1}{z}\right)$ |
| $\mathcal{T}$ | $\left(\frac{1+\epsilon}{2}\right)\left(z_1 + \frac{z_2}{z_1} + \frac{1}{z_2} + z_2 + \frac{z_1}{z_2} + \frac{1}{z_1}\right)$ |

**Table 6**. Character for the $F \oplus \overline{F}$ representation of $\widetilde{\mathrm{SU}}(3)$ for various fugacity subgroups.

| Group | Adjoint Character |
|---|---|
| $T$ | $2 + \frac{z_1^2}{z_2} + z_1 z_2 + \frac{z_2^2}{z_1} + \frac{z_1}{z_2^2} + \frac{1}{z_1 z_2} + \frac{z_2}{z_1^2}$ |
| $T^+$ | $\left(\frac{1+\epsilon}{2}\right)\left(2 + \frac{z_1^2}{z_2} + \frac{z_2^2}{z_1} + \frac{z_1}{z_2^2} + \frac{z_2}{z_1^2}\right) + \epsilon\left(z_1 z_2 + \frac{1}{z_1 z_2}\right)$ |
| $T^0$ | $(1 + \epsilon)\left(1 + z + z^{-1}\right) + \epsilon\left(z^2 + z^{-2}\right)$ |
| $\mathcal{T}$ | $\left(\frac{1+\epsilon}{2}\right)\left(\frac{z_1^2}{z_2} + z_1 z_2 + \frac{z_2^2}{z_1} + \frac{z_1}{z_2^2} + \frac{1}{z_1 z_2} + \frac{z_2}{z_1^2}\right) + 2\epsilon$ |

**Table 7**. Character for the adjoint representation of $\widetilde{\mathrm{SU}}(3)$ for various fugacity subgroups.

where we have defined

$$\psi(z_1, z_2, \epsilon) = \left(D\left(z_1, \frac{z_2}{z_1}, \frac{1}{z_2}\right), \epsilon\right). \tag{A.24}$$

The product rules give

$$\psi(z_1, z_2, \epsilon) \cdot \psi(y_1, y_2, \eta) = \begin{cases} \psi(z_1 y_1, z_2 y_2, \epsilon\eta) & \text{if } \epsilon = 1 \\ \psi(z_1 y_1^{-1}, z_2 y_2^{-1}, \epsilon\eta) & \text{if } \epsilon = -1 \ . \end{cases} \tag{A.25}$$

This means that $\psi$ is an injective group morphism $\mathrm{U}(1)^2 \rtimes \mathbb{Z}_2 \to \widetilde{\mathrm{SU}}(3)$ where the semidirect product $\mathrm{U}(1)^2 \rtimes \mathbb{Z}_2$ is defined by

$$(z_1, z_2, \epsilon) \cdot (y_1, y_2, \eta) = \begin{cases} (z_1 y_1, z_2 y_2, \epsilon\eta) & \text{if } \epsilon = 1 \\ (z_1 y_1^{-1}, z_2 y_2^{-1}, \epsilon\eta) & \text{if } \epsilon = -1 \end{cases} \tag{A.26}$$

This is a different from (A.21). In (A.21) the $\mathbb{Z}_2$ acts on $\mathrm{U}(1)^2$ by permuting the two factors, while here is inverts elements in both factors and preserves the order. The subgroup $\mathcal{T}$ is not a Cartan subgroup, as it does not preserve the fundamental Weyl chamber. However its matrices are all diagonal, and therefore is well suited for deriving branching rules.

### A.3 Characters

A representation of $\widetilde{\mathrm{SU}}(N)$ is a vector space $V$ with a group morphism $\rho : \widetilde{\mathrm{SU}}(N) \to GL(V)$. Picking a basis for $V$, a finite dimensional representation is given by matrices $\rho(g, 1)$ and $\rho(g, -1)$ for each $g \in \mathrm{SU}(N)$, satisfying the product rule $\rho(g, \epsilon)\rho(g', \epsilon') = \rho(g\Theta_\epsilon(g'), \epsilon\epsilon')$. The character $\chi$ of this representation is the trace of these matrices: $\chi(g, \epsilon) = \mathrm{Tr}(\rho(g, \epsilon))$. Note that using (A.10) we have $\chi(g, \epsilon) = \chi(g'\Theta_{\epsilon'}(g)\Theta_\epsilon(g')^{-1}, \epsilon)$ for any $g$, $g'$, $\epsilon$ and $\epsilon'$.

In particular

$$\chi(g, 1) = \chi(hgh^{-1}, 1) \tag{A.27}$$

$$\chi(g, -1) = \chi(hg\Theta_{-1}(h)^{-1}, -1) \tag{A.28}$$

$$\chi(g, 1) = \chi(h\Theta_{-1}(g)h^{-1}, 1) \tag{A.29}$$

$$\chi(g, -1) = \chi(h\Theta_{-1}(g)\Theta_{-1}(h)^{-1}, -1) \tag{A.30}$$

In order to express these characters we pick a diagonal form $g = \mathrm{Diag}(z_1, \ldots, z_N)$, as explained in the previous subsection. Let's see what the constraints above tell us about the function $\chi(z_i, \epsilon)$, taking the case of type I to illustrate. The third lines says that the character for $\epsilon = 1$ is invariant under $z \to z^{-1}$. The second line says that $\chi(g, -1) = \chi(hgh^T, -1)$ for any $h \in \mathrm{SU}(N)$. In particular for $h = \mathrm{Diag}(h_i)$ with $|h_i| = 1$ this gives $\chi(z_i, -1) = \chi(h_i^2 z_i, -1)$. In other words, $\chi(z_i, -1)$ can not depend on the $z_i$ at all! Therefore it is a pure number that can be evaluated for $z_i = 1$.

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
