# Peer review of "Discrete gauging and Hasse diagrams"

_SciPost Physics_

## Round 1 · Referee Report · Anonymous (Referee 1) · 2021-6-21

Strengths

  1. This article contains several new interesting results.
  2. The presentation is clear and readable.
  3. The proposals were tested and checked using a number of exact computations.

Report

This article investigates the 4d $\mathcal{N}=2$ supersymmetric QCD with a disconnected gauge group that is a semidirect product between $SU(N)$ and the charge conjugation symmetry. There are two types of such products. The authors studied the structure of the Higgs branch of the theories of both types as a symplectic singularity using Hasse diagrams. The magnetic quivers for one of the types were proposed. It turned out that they are wreathed quivers, recently discovered by one of the authors along with other collaborators. In the referee's opinion, the article deserves publication after minor corrections and improvements.

Requested changes

  1. The authors proposed the Hasse diagrams in Figures 3 and 5 for the case of $N_f \geq N_c$. The magnetic quivers for such a case were also depicted in Figures 7 and 8. I would like to ask the authors to comment on the case of $N_f < N_c$. What would be the corresponding Hasse diagrams and magnetic quivers? Can these be determined using the technique, such as in [arXiv:1909.00667]?
  2. The authors presented the unrefined Hilbert series in (4.19). How can this be refined? Can one turn on the relevant fugacities in (4.1) and (4.4)? The authors should demonstrate how to do so at least to a few orders in the power series of the Hilbert series. Moreover, are there any fugacities associated with discrete symmetries one can turn on?
  3. Some minor typos on page 20: $\widetilde{SU}_I(N)$ should be changed to $\widetilde{SU}(N)_I$ for consistency.

  • validity: high
  • significance: high
  • originality: high
  • clarity: high
  • formatting: excellent
  • grammar: excellent

Author:  Antoine Bourget  on 2021-07-07  [id 1552]

(in reply to Report 1 on 2021-06-21)

We would like to thank the referee for their precise reading of our paper.

We have addressed the requested changes as follows:

  1. We decided to restrict to $N_f \geq N_c$ as the case $N_f < N_c$ is much more involved, as is already visible in SQCD with $\mathrm{SU}$ gauge groups. One difficulty comes from the fact that the Higgs branch may be a union of several hyperK\"ahler cones with non-trivial intersections. In addition, non complete Higgsing gives rise in some cases to nilpotent operators in the Higgs chiral ring which makes it difficult to match with a 3d $\mathcal{N}=4$ Coulomb branch ring. Therefore although we are able to construct a putative Hasse diagram using classical Higgsing, we don't know how to extend magnetic quivers below that bound. In order to apply the techniques of [arXiv:1909.00667], we would need to have a brane construction, which is still missing. We have added footnote 5 in the text to explain these points.

  2. It is indeed possible to refine the Hilbert series and we did it explicitly in the added equations (4.20)-(4.22). We don't expect there to be remaining discrete fugacities to turn on. (However it should be possible to consider the CB Hilbert series for the initial quiver (4.5) and use $C_3$ and $\mathbb{Z}_2$ fugacities in order to deduce the Hilbert series for the folded quiver; we did not mention this in our paper as we focus on the wreathed quiver).

  3. The typo is corrected.

---

## Round 1 · Referee Report · Anonymous (Referee 3) · 2021-6-28

Strengths

1 - Novel results in a topic of current interest
2 - High level of sophistication
3 - Cross-checks of the main claims
4 - Clarity of exposition and self-containedness
5 - Limitations and open problems clearly spelled out.

Weaknesses

1 - No discussion or tests of whether the wreathed magnetic quivers are 3d mirrors of the $\widetilde{SU}(N)$ SQCD theories.

Report

This is an interesting, timely and well written paper on the Higgs branch of SQCD with principally extended $\widetilde{SU}(N)$ gauge group and eight supercharges. The authors derive the Hasse diagram which encodes the stratification of the Higgs branch from a detailed study of the Higgsing patterns. They also conjecture that the corresponding magnetic quivers are recently introduced wreathed quivers and provide detailed checks of this conjecture using Hilbert series.

Challenges in finding magnetic quivers for $\widetilde{SU}(N)_{II}$ theories remain, as the author clearly explain. The article also contains interesting remarks on the relations among wreathed quivers, folded quivers, discrete gaugings and twisted compactifications, which help clarify the status of the subject.

I believe that the article is worthy of publication subject to a minor revision to address a few minor points, which I list below.

Requested changes

1 - Do the authors expect that the proposed magnetic quivers are also the 3d mirrors of the principally extended gauge theories? Please include a discussion. 2 - Transverse slices to leaves of the Higgs branch are deduced from the number of singlets in decompositions such as (3.4), which however only counts the dimension of the slice. For the benefit of uninitiated readers, please explain briefly how to discriminate among different slices of equal dimension. 3 - A few typos need to be corrected:

p 2 last para: no trivial -> non-trivial; p 3 last para: missing of in first line; no-trivial -> non-trivial; p 5, first line of section 2.1: setup -> set up; p 6 footnote 3: $"$ -> $``$ before pseudo; p 20 fifth line of section 4: Coulomb branch of the is missing before wreathed quivers; p 31 fourth line: $SU(3)$ or $\widetilde{SU}(3)$?

  • validity: high
  • significance: high
  • originality: high
  • clarity: high
  • formatting: excellent
  • grammar: excellent

Author:  Antoine Bourget  on 2021-07-07  [id 1551]

(in reply to Report 3 on 2021-06-28)

We would like to thank the referee for their precise reading of our paper.

We have addressed the requested changes as follows:

1- We didn't make a claim about the wreathed quivers to be 3d mirrors of $\widetilde{SU}$ gauge theories as this would require an in-depth analysis of monopole operators for that gauge group in order to match the 3d Coulomb branch of that theory with the Higgs branch of the wreathed quiver. We have added a paragraph in Section 4.3 to explain this point.

2- Slices of equal dimension can be distinguished by the dimension of the gauge group of the theory defining the slice (1, 0 or 3 for the slices present in this paper). We have included a discussion of that point at the beginning of Section 3.

3- The typos are corrected.

Anonymous on 2021-07-07  [id 1553]

(in reply to Antoine Bourget on 2021-07-07 [id 1551])
Category:
remark

I am satisfied with the minor corrections. The paper is ready for publication as far as I am concerned.

---

## Round 1 · Referee Report · Anonymous (Referee 2) · 2021-6-28

Strengths

1- Studies in detail the representations and their branchings for the $\widetilde{SU}_{I,II}$ groups

2- Proposes a magnetic quiver for the $\tilde{SU}_I$ groups, and provides, as a strong check of the conjecture, evidence for the matching of the corresponding Hilbert series

Weaknesses

1- No real weakness

Report

In this paper the authors study theories based on disconnected gauge groups obtained as principal extensions of unitary groups, which correspond to gauging charge conjugation. These groups come in two varieties dubbed $\widetilde{SU}(N)_{I,II}$, of which the $\widetilde{SU}(N)_{II}$ requires $N$ to be even.

The paper has two parts: in the first part some relevant representations and their branching rules are studied. Using this, the Hasse diagrams for Higgs branches of theories based on these groups with fundamental matter is constructed. In the second part the magnetic quiver for the $\widetilde{SU}(N)_I$ with fundamental matter is constructed. It turns out to be a wreathed version of the magnetic quiver for the ``parent" theory with $SU(N)$ group.

I think that the paper is an interesting technical development and thus I recommend it for publication.

Requested changes

1- Since quiver wreathing plays a very relevant role, a few more words on quiver wreathing would be acknowledged to make the paper more self-contained.

2- The identification of the wreathed quiver as magnetic dual to the $\widetilde{SU}$ theories with fundamental matter is very interesting, and the fact that the Higgs branch Hilbert series is reproduced is a strong test. However, as the authors point, this doesn't necessarily imply that the magnetic quiver captures the same symplectic singularity. Nevertheless, below the three points $\alpha)-\gamma)$ on page 24 the authors conjectured that they are indeed isomorphic. It would be good if the authors could elaborate a bit more on the motivation of this conjecture.

  • validity: high
  • significance: high
  • originality: high
  • clarity: high
  • formatting: good
  • grammar: excellent

Author:  Antoine Bourget  on 2021-07-07  [id 1554]

(in reply to Report 2 on 2021-06-28)

We would like to thank the referee for their precise reading of our paper.

We have addressed the requested changes as follows:

1- We have added a paragraph at the beginning of Section 4.1 giving the detailed definition of wreathed products and how this is used to construct wreathed quivers.

2- The conjecture that the symplectic singularities are indeed identical comes from the fact that in addition to the Hilbert series computation, the magnetic quiver comes in a natural way (wreathing) from a standard unitary quiver which is known to be the 3d mirror of the 3d compactification of SU(3)+6F 4d SQCD. Moreover, the $\mathbb{Z}_2$ which is used in the wreathing operation is identified in [1605.08337] with charge conjugation, which is precisely what is gauged in the $\widetilde{SU}$ gauge theory. We have added footnote 7 to explain this.

Anonymous on 2021-07-28  [id 1625]

(in reply to Antoine Bourget on 2021-07-07 [id 1554])

The claim made in the paper can indeed be seen to be equivalent to the claim that the three operations 1) Gauging charge conjugation 2) Taking a magnetic quiver 3) Wreathing the magnetic quiver form a commuting square. In words, this says that starting from the theory SU(3)+6F one can either

  • Construct a magnetic quiver, then wreath it by the appropriate Z_2 and construct the Coulomb branch

or

  • Gauge charge conjugation and compute the Higgs branch

both give the same space.

The fact that both construction give the same space is comforted by the Hilbert series computation given in the paper, but it is not a full proof. This is the reason why we only give step (gamma) on page 25 of our paper as a conjecture. It would indeed be interesting to perform further checks such as the space of deformations of both singularities. We decided to keep this for future work.

Anonymous on 2021-07-09  [id 1559]

(in reply to Antoine Bourget on 2021-07-07 [id 1554])

That the Z_2 used for wreathing stands for charge conjugation is a necessary condition for the two singularities to be the same, but it seems a bit short as evidence that the singularities are the same. In fact, without 1605.08337 the present paper would serve to identify the wreathing Z_2 as the magnetic avatar charge conjugation. With 1605.08337, the present paper simply makes the correct choice of that Z_2. But identifying both singularities would require more. For instance, is it expected that the space of deformations is the same in both the original theory and in the magnetic one?

The authors raise that the starting point for gauging charge conjugation/wreathing is a mirror pair. Is it being implicitly conjectured that gauging charge conjugation/wreathing “commutes” with mirror symmetry, so that the resulting theories are truly mirror?

---

## Editorial Decision

resubmitted